# Novel Siderophore Cephalosporin and Combinations of Cephalosporins with β-Lactamase Inhibitors as an Advancement in Treatment of Ventilator-Associated Pneumonia

**DOI:** 10.3390/antibiotics13050445

**Published:** 2024-05-14

**Authors:** Szymon Viscardi, Ewa Topola, Jakub Sobieraj, Anna Duda-Madej

**Affiliations:** 1Faculty of Medicine, Wroclaw Medical University, Ludwika Pasteura 1, 50-367 Wrocław, Poland; ewa.topola@student.umw.edu.pl (E.T.); jakub.sobieraj@student.umw.edu.pl (J.S.); 2Department of Microbiology, Faculty of Medicine, Wroclaw Medical University, Chałubińskiego 4, 50-368 Wrocław, Poland

**Keywords:** cefoperazone-sulbactam, ceftolozane-tazobactam, ceftazidime-avibactam, cefiderocol, ventilator-associated pneumonia, hospital-associated pneumonia, nosocomial infections, multidrug-resistant, Gram-negative bacteria

## Abstract

In an era of increasing antibiotic resistance among pathogens, the treatment options for infectious diseases are diminishing. One of the clinical groups especially vulnerable to this threat are patients who are hospitalized in intensive care units due to ventilator-associated pneumonia caused by multidrug-resistant/extensively drug-resistant Gram-negative bacteria. In order to prevent the exhaustion of therapeutic options for this life-threatening condition, there is an urgent need for new pharmaceuticals. Novel β-lactam antibiotics, including combinations of cephalosporins with β-lactamase inhibitors, are proposed as a solution to this escalating problem. The unique mechanism of action, distinctive to this new group of siderophore cephalosporins, can overcome multidrug resistance, which is raising high expectations. In this review, we present the summarized results of clinical trials, in vitro studies, and case studies on the therapeutic efficacy of cefoperazone-sulbactam, ceftolozane-tazobactam, ceftazidime-avibactam, and cefiderocol in the treatment of ventilator-associated pneumonia. We demonstrate that treatment strategies based on siderophore cephalosporins and combinations of β-lactams with β-lactamases inhibitors show comparable or higher clinical efficacy than those used with classic pharmaceuticals, like carbapenems, colistin, or tigecycline, and are often associated with a lower risk of adverse events.

## 1. Introduction

Ventilator-associated pneumonia (VAP) is a major health problem occurring in intensive care units (ICUs), the importance of which may increase due to arising antibiotic resistance. VAP is defined as a pneumonia that arises more than 48–72 h after endotracheal intubation [1]. Half of all cases of hospital-acquired pneumonia (HAP) are due to ventilation, with 9% to 27% of intubated patients at risk of developing this condition and with an event risk of around 1.2 to 8.5 per 1000 ventilation days. The excess cost of VAP treatment is estimated to be over USD 40,000 per patient [1,2]. However, the prevalence may vary depending on the country; a large retrospective study in the USA estimated that 1.81% of mechanically ventilated patients developed VAP [3], while in a study conducted in India, it reached 34% of patients [4]. Also, the mortality of VAP may be different and has been reported to be up to 50%, but this may be distorted by confounding factors, as patients admitted to the ICU have initially serious conditions. Therefore, the attributable mortality is estimated to be 9% to 13% [5], which still makes VAP an important clinical issue.

VAP is usually a bacterial infection. The main pathogens responsible for 80% of ventilator-associated bacterial pneumonia (VABP) are ESKAPE group pathogens: *Enterococcus faecium*, *Staphylococcus aureus*, *Klebsiella pneumoniae*, *Acinetobacter baumannii*, *Pseudomonas aeruginosa*, and *Enterobacter* spp. [6]. Viruses as a sole cause of VAP in the ICU are rare [7]. The most frequently detected are HSV-1 (*Herpes simplex* virus-1) infections (31% of cases) [7] and rhinovirus, influenza A virus and cytomegalovirus (CMV) (5.1% of patients with VAP) [7,8]. However, acute respiratory distress syndrome (ARDS) due to SARS-CoV2 (severe acute respiratory syndrome coronavirus-2) infection can lead to bacterial superinfection and VABP in 21–64% of patients [9]. Even though detection of *Candida* spp. respiratory tract colonization in patients with VAP is associated with higher mortality (45.45% vs. 28.67% in VAP without confirmed colonization, *p* < 0.05), the role of fungi in the pathogenesis of VAP is controversial [10]. Therefore, although other pathogens may contribute to the progress and outcome of VAP, bacteria remain the most important etiological factor.

Because of the fact that bacteria are the main etiological agent associated with VAP, the problem of drug resistance is a real-life risk for these patients. Around 50% of antibiotics used in ICUs are administered because of VABP, but they are often ineffective due to the high proportion of resistant bacteria [11]. They can exhibit a panel of resistance to antimicrobial categories (AMCs): (i) multidrug-resistant (MDR)—nonsusceptible to at least one agent in three or more AMCs; (ii) extensively drug-resistant (XDR)—nonsusceptible to at least one agent in all but two or fewer AMCs; and (iii) pan-drug-resistant (PDR)—nonsusceptible to all antimicrobial agents [12]. Meta-analyses performed showed that *P. aeruginosa* MDR was the etiologic agent of VABP in 33% of cases [13]. However, *A. baumannii* isolates showed resistance rates of 13.3%, 68.3%, and 18.3% for MDR, XDR, and PDR, respectively [14]. In such cases, empiric antibiotic therapy may not be sufficient to completely eradicate the bacterial agent. According to 2016 guidelines from the American Society of Infectious Diseases/American Thoracic Society, a local epidemiological panel should be considered before initiating treatment. However, due to the easy distribution of resistance genes in the hospital environment, appropriate antibiotics may still be limited [15]. In the calculation of this problem, new therapeutic options from the group of β-lactams are promising. There are reports of the high effectiveness of these preparations against carbapenem-resistant *Enterobacterales*, which are an increasing danger for patients hospitalized in the ICU [16,17]. In turn, therapies with cephalosporins, β-lactam antibiotics, whose mechanism of action is to inhibit bacterial cell wall synthesis, are showing promising results. However, as a result of evolution and drug abuse, bacteria have developed resistance to β-lactams, mainly through the production of β-lactamases that inactivate these antibiotics. Recent drugs can combine these antibiotics with β-lactamase inhibitors, reducing bacterial resistance to therapy. These combinations include ceftazidime-avibactam (CAZ-AVI), cefoperazone-sulbactam (CFP-SBT), and ceftolozane-tazobactam (CEF-TAZ). One group of cephalosporins that may contribute to more effective treatment of VABP are the newly developed siderophore cephalosporins, such as cefiderocol (CFD). Table 1 summarizes the abilities of formulations to overcome multidrug resistance among bacteria according to the Ambler classification.

The aim of this study was to evaluate the therapeutic potential of novel drugs in VABP treatment. In this review, we carefully compiled the data on the clinical efficiency of the mentioned cephalosporins combinations with β-lactamase inhibitors and cefiderocol, as well as their in vitro activity against clinical isolates derived from patients with confirmed VABP. The collection of this information is not only important for researchers focused on β-lactams, but, more importantly, we believe that our review will help clinicians offer accurate therapeutic options for treating patients in the ICU.

## 2. Characteristics of Cefoperazone-Sulbactam

CFP-SBT is a combination of the antibiotic cefoperazone, a III^rd^-generation cephalosporin, and a class A β-lactamase inhibitor, sulbactam [18]. It has significant activity against Gram-negative bacteria, e.g., *Enterobacterales*, *Acinetobacter* spp., and *P. aeruginosa* [24]. This combination increases the effectiveness of the antibiotic against MDR bacteria, except *P. aeruginosa* resistant to carbapenems. CFP-SBT is used at antibiotic-to-inhibitor ratios of 2:1, 1:1, and 1:2, which lead to increasing reductions in the MIC (minimal inhibitory concentration) [25]. Moreover, bacterial resistance to the combination in the 1:1 ratio is lower compared to the tested combination in the 2:1 ratio [26]. Sulbactam addition in the 1:1 ratio reduced the MIC_90_ > 128 mg/L for *Enterobacter cloacae* (MIC_90_ = 32 mg/L; MIC_90_—the lowest concentration of the antibiotic at which 90% of the isolates were inhibited); *Serratia marcescens* (MIC_90_ = 32 mg/L), *K. pneumoniae* (ESβL: 32 mg/L; non-ESβL: 4 mg/L); and *A. baumannii* (imipenem resistant: 32 mg/L; imipenem susceptible: 16 mg/L). The 2:1 ratio also reduced the bacterial MIC [27]. In vitro tests showed that the tested combination is active against *Acinetobacter* spp., with an MIC value of 1.0 µg/mL [28]. Potential uses of this drug include infections of the lower and upper respiratory systems, infections of the lower and upper urinary systems, intra-abdominal infections, sepsis, skin and soft tissue infections, joint and bone infections, and bacterial infections of the genitals, e.g., gonorrhea. The indications listed are part of Pfizer’s guidelines. The combination should be administered in a 1:1 or 1:2 ratio. For a 1:1 ratio, 1–2 g of sulbactam and 1–2 g of cefoperazone are used, and for a 1:2 ratio, 0.5–1 g of sulbactam and 1–2 g of cefoperazone. Doses should be administered to the patient every 12 h. The described mode of administration is intended for adults and is part of Pfizer’s guidelines. A very important issue is the degree of drug penetration into the lungs. A group of patients (121 people) with infections caused by *A. baumannii* obtained CFP-SBT at a dose of 3 g every 8 h, but the effectiveness of the drug decreased over time. It is possible to administer a higher dose (4 g), which may improve the quality of treatment [29]. Adverse events connected with CFP-SBT treatment include nephrotoxicity, thrombocytopenia, leukopenia, increased liver enzyme activity, prolonged prothrombin time (PT), and increased international normalized ratio (INR) [30]. However, the most common side effect is a rash (which occurs in 10.1% of patients), as well as vomiting (4.4%) [31].


**Cefoperazone**


Cefoperazone (CFP)—(6R,7R)-7-[[(2R)-2-[(4-ethyl-2,3-dioxopiperazine-1-carbonyl) amino]-2-(4-hydroxyphenyl)acetyl]amino]-3-[(1-methyltetrazol-5-yl)sulfanylmethyl]-8-oxo-5-thia-1-azabicyclo[4.2.0]oct-2-ene-2-carboxylic acid (IUPAC name according to PubChem) is a semisynthetic, III^rd^-generation cephalosporin. It is an antibiotic intended for intramuscular or intravenous (i.v.) infusions. In contrast to I^st^- and II^nd^-generation cephalosporins, it is active against *P. aeruginosa* and shows greater stability to hydrolysis by bacterial enzymes. Its action also covers Gram-positive and Gram-negative bacteria, including *Enterobacteriaceae* and anaerobes [32]. The combination of sulbactam with this cephalosporine expands its spectrum to, among others, *Bacteroides* spp. and *Acinetobacter* spp. [33]. The structural model of cefoperazone is shown in Figure 1A.


**Sulbactam**


Sulbactam (SBT)—(2S,5R)-3,3-dimethyl-4,4,7-trioxo-4lambda6-thia-1-azabicyclo [3.2.0] heptane-2-carboxylic acid (IUPAC name according to PubChem), acts on bacteria with an ESβL-resistance mechanism (such as *E. coli* or *K. pneumoniae*) and AmpC (cephalosporinases encoded on the chromosomes of many of the *Enterobacteriaceae*), as well as against *Acinetobacter* resistant to carbapenems [18,26]. On the molecular level, SBT contains a β-lactam ring in its structure. Sulbactam presented significant activity on bacterial enzymes transferred by plasmids [33]. Studies on *A. baumannii* showed that sulbactam also inhibits penicillin-binding proteins (PBPs) (mainly PBP1 and PBP3), which are key proteins in bacterial cell wall synthesis. This action leads to the death of bacterial cells [34,35]. Resistance to sulbactam is influenced by TEM β-lactamases, OXA-23 enzymes, and ADC (*Acinetobacter*-derived cephalosporinase). Mutations in PBPs can significantly increase sulbactam’s MIC [29]. Importantly, SBT has an impressive effect on plasmid-mediated β-lactamases in contrast to CFP alone [28]. The structural model of sulbactam is shown in Figure 1B.

### 2.1. Clinical Efficiency of Cefoperazone-Sulbactam

#### 2.1.1. In Vitro Studies

The Namiduru et al. study evaluated the in vitro activity of available antibiotics against 230 clinical strains of bacteria isolated from patients with confirmed VABP between 2001 and 2003. The predominant pathogens were *P. aeruginosa* (33.9%), *S. aureus* (30%), *A. baumannii* (26.1%), and *Enterobacter* spp. (4.3%). The study showed that the highest efficacy against *P. aeruginosa* and *A. baumannii* isolates was obtained by CFP-SBT (80.7% vs. 80%, respectively) and imipenem (*A. baumannii* 83.3%) [36]. Similar results come from the Xia et al. study conducted in a population of pediatric patients (N = 94) hospitalized in an ICU with a confirmed diagnosis of VABP. Dominant pathogens isolated from the respiratory tract were identified as MDR isolates of *K. pneumoniae* and *A. baumannii* (78%). The highest activity against *A. baumannii* was detected (similarly to the Namiduru et al. study above) in the case of CFP-SBT and imipenem [37]. Similar reports to those from Xia et al. and Namiduru et al. come from the Capoor et al. study, in which sensitivity to antibiotics was assessed among 128 clinical isolates (derived from patients diagnosed with VABP). The predominant isolated pathogens were *Acinetobacter calcoaceticus-Acinetobacter baumannii* complex. CFP-SBT was found to be the most effective against the pathogen, with 95.6% of strains being effectively treated with it, along with MER and PIP-TAZ [38].

Another study evaluated the in vitro susceptibility to CFP separately and in combination with SBT against bacterial isolates from patients hospitalized (N = 1796 Gram (−) isolates and 476 Gram (+)) in several hospitals in Taiwan in 2012. The most frequently isolated pathogens identified as etiological factors of VABP/HAP in ICU wards were MSSA (methicillin-susceptible *S. aureus*) (10.65%), *E. coli* (11%), *K. pneumoniae* (10.17%), *A. baumannii* (16.51%), *P. aeruginosa* (7.83%), and *Stenotrophomonas maltophila* (3.83%). The study showed a higher activity of the combination of CFP-SBT over CFP alone for MSSA (MIC_50_ = 2 µg/mL vs. 4 μg/mL; MIC_50_—the lowest concentration of the antibiotic at which 50% of the isolates were inhibited) for Gram (+) pathogens. Pathogens of the ESKAPE group were also characterized by a higher sensitivity to the combination with the inhibitor *K. pneumoniae* and *P. aeruginosa* (MIC_90_ = 32 vs. 128/mL), *A. baumannii* (MIC_90_ = 64 vs. >128 µg/mL), and *S. maltophila* (MIC_50_ = 64 vs. 128 µg/mL). The CFP-SBT showed elevated activities against *K. pneumoniae* and *E. coli* ESβL strains, as well as against CRAB and CRPA (carbapenem-resistant *P. aeruginosa*) in comparison to CFP alone [39].

In another study, Huang et al. evaluated the in vitro efficacy of combinations of azithromycin (AZT) with CFP-SBT for clinical isolates of MDR *P. aeruginosa* collected from 151 ICU patients diagnosed with VABP. The strains were characterized by a high degree of resistance to, among others, aztreonam (ATM) or imipenem. The study included a total of four schemes of empirical therapy containing CFP-SBT, as follows: CFP-SBT + AZT, CFP-SBT + amikacin (AKC), CFP-SBT + levofloxacin (LEV), and CFP-SBT + AZT + LEV/AKC combination. The highest effectiveness was achieved by combining CFP-STB with AKC (70%) and AZT (60%). The combination with AZT allowed for a decrease in the MIC_50_ for CFP-SBT from 64 to 16 μg/mL and MIC_90_ from 128 to 64 μg/mL. It has been shown that both the combinations of the preparation with AZT and AZT + AKC provide an additive effect. The results indicate that the empirically tested combination of BL + BLI (β-lactam + β-lactamase inhibitors) with AZT may have potential benefits in treating VABP caused by *P. aeruginosa* [40].

Xia et al. evaluated the in vitro efficacy of CFP-SBT therapy in combination with other antibiotics against CRAB isolates derived from patients hospitalized in the ICU (N = 71, of which 45 were subjected to mechanical ventilation). Patients qualified for the study, due to the lack of a possibility of receiving polymyxin therapy (renal failure), were randomized into the following two groups: those receiving and not receiving a CPF-SBT therapy-based scheme. A higher 30-day survival ratio was demonstrated in the CFP-SBT-receiving population (96.4% vs. 73.3%). In vitro studies showed a high percentage of expression of the following *A. baumannii* carbapenemases: OXA-23 and OXA-51. The evaluation of antibiotic sensitivity involved choosing multiple combinations with additive/synergistic properties, and the combination of CFP-SBT + MER was revealed to be the most active connection (MIC_50_ = 16 µg/mL; MIC_90_ = 64 µg/mL) [41]. The Sader et al. study evaluated the in vitro susceptibility of *Enterobacteriaceae* isolates (including *P. aeruginosa* (3818) and *A. baumannii* (1310)) to CFP-SBT. A total of 28.3% of all isolates were collected from patients diagnosed with HAP/VABP. The susceptibility profile of *P. aeruginosa* to the preparation was defined in the range of 59.5–83%, of which the highest activity was recorded in relation to strains from Western Europe and Latin America (83%). In turn, the sensitivity of *A. baumannii* to the preparation was relatively lower at 43–73.8% [24]. A summary of the in vitro antimicrobial activity of cefoperazone-sulbactam is presented in Table 2.

#### 2.1.2. Clinical Trials

The randomized noninferiority clinical trial was conducted to evaluate the therapeutic efficacy of CFP-SBT vs. cefepime (CPM) for the treatment of HAP and healthcare-associated pneumonia (HCAP). Ultimately, 166 patients were randomized into the following two groups: those treated with CFP-SBT (N = 79) and with CPM (N = 87). Patients were assessed twice during the test-of-cure (TOC) in the following phases: intention-to-treat (ITT) and per protocol (PP). Clinical success has been shown for patients receiving CFP-SBT and CPM treatment after approximately 10 days in both the ITT and PP analyses. In turn, the percentages of clinical cures with CFP-SBT therapy obtained better parameters in contrast to CPM, as follows: clinical cure at ITT analysis = 73.1% vs. 56.8% and clinical cure at PP analysis = 74.2% vs. 56.8%. Among the sputum-isolated pathogens obtained from patients treated with CFP-SBT, the presences of *P. aeruginosa* (100% effective therapy) and *A. baumannii* (75% effectiveness) were found. Both formulations achieved a similar adverse event profile (CFP-SBT 58/79, of which 20 were directly drug related; CPM 58/87, of which 15 were directly drug related). It was also shown that in the case of patients suffering from HAP, CFP-SBT showed a lower effectiveness than CPM in terms of the percentage of clinical cure (~43% vs. ~88%). The study proves that the preparation is noninferior to CPM activity and is an interesting option for HAP/VABP therapy [43].

Another study compared the efficacy and safety profile of CFP-SBT and piperacillin-tazobactam (PIP-TAZ) in elderly patients hospitalized due to, among others, HAP/VABP. Patients were randomly assigned into the following two groups: those receiving PIP-TAZ (N = 150) and those receiving CFP-SBT (N = 167); the most commonly isolated pathogens were *K. pneumoniae*, *P. aeruginosa*, and *A. baumannii*. Clinical cure was achieved in both groups at 85 vs. 81%, respectively and clinical failure at 14 and 17%, respectively. The study also revealed the in-hospital mortality ratio (including pneumonia-related mortality), it was 40/167 (including 24 due to pneumonia) and 32/150 (of which 14 related to pneumonia) for CFP-SBT and PIP-TAZ, respectively. In the scope of TEAE (treatment-emergent adverse event), both preparations obtained comparable outcomes. Diarrhea was the predominant type in the PIP-TAZ group (N = 2). In turn, patients treated with CFP-SBT developed hepatitis, rash, and hemorrhage (in total, N = 3) [42]. The Chen et al. study compared the effectiveness of therapy, among others, in VABP in patients treated with CFP-SBT (N = 37) or PIP-TAZ (N = 28). The most commonly isolated pathogens in both groups were *A. baumannii*, *P. aeruginosa*, and *P. aeruginosa*. In the study, similar results were obtained for clinical cure in patients hospitalized due to VABP for both CFP-SBT and PIP-TAZ at 78.4 vs. 71.4% and treatment failures at 16.2 vs. 28.6%. However, the mortality ratio due to pneumonia was higher for CFP-SBT therapy vs. PIP-TAZ (13.5 vs. 3.6%). Nevertheless, the results of the study clearly show that cefoperazone sulbactam is equally effective in VABP therapy as its comparator, PIP-TAZ [44].

The Guclu et al. study evaluated the efficacy of CFP-SBT and PIP-TAZ therapy in the treatment of nosocomial infections in patients hospitalized at the ICU of Sakarya University Training and Research Hospital in the years 2017–2018. Patients were randomly assigned to the following two groups: those receiving empirical therapy based on CFP-SBT or PIP-TAZ (number in each group, N = 154). The percentages of patients hospitalized for HAP/VABP in both groups were 17.5% and 22.5%, respectively. Pathogens classified as MDR accounted for more than 70% of all etiological factors of diseases, the predominant species included *K. pneumoniae*, *E. coli*, *P. aeruginosa*, and *A. baumannii*. Both preparations had similar therapeutic parameters, as follows: treatment success—50% vs. 51.2%; 14-day mortality ratio—29.2% vs. 35%; 28-day mortality ratio—46.1% vs. 42.8%; and TEAEs frequency—50.6% vs. 46.1%. The most common adverse events were increased INR value, thrombocytopenia, hepatotoxicity, and nephrotoxicity. As it was demonstrated by other studies in this field [42,44], CFP-SBT had a noninferior activity profile compared to PIP-TAZ in the empirical therapy of HAP/VABP [30].

Kara et al. evaluated the efficacy of colistin (COL) monotherapy and its combinations with other antibiotics in VABP therapy. Between 2009 and 2014, 134 ICU patients were enrolled in the study, and *A. baumannii* was the most common etiological factor for VABP. One of the tested combinations was the combination of high doses of CFP-SBT with COL, which resulted in a relatively low effectiveness (median duration of therapy: 6 days; treatment failure: >70%; and mortality ratio: 15/17 of the patients); however, the results were comparable to the effectiveness of the combination of COL + tigecycline (TGC) or COL + ampicillin-sulbactam [45]. The Qin et al. study evaluated the effect of a high-dose combination of CFP-SBT with TGC on the efficacy of VABP therapy with the etiology of XDR *A. baumannii*. Patients were randomized into the following two groups: receiving a combination of both drugs and receiving TGC as monotherapy. The study showed a higher efficacy of CFP-SBT + TGC scheme vs. TGS therapy separately (85.7% vs. 47.6% clinical success ratio) and also demonstrated the ability of CFP-SBT to lower the MIC of TGC. In both study groups, there were no significant differences in the incidence of TEAE [46]. Another study presented interesting reports on the effectiveness of VABP therapy of etiology of carbapenem-resistant *A. baumannii* (CRAB) using CFP-SBT. A total of 80 patients were enrolled in the study and randomized into the following two groups: those who received CFP-SBT as an adjunct to COL, meropenem (MER), or TGC (N = 52); those who received BAT (best available therapy; drug scheme without CFP-SBT), N = 38. Significantly, differences in mortality were found in both groups, as follows: 14-day mortality ratio—17 vs. 39%; 30-day mortality ratio—35 vs. 61%; in-hospital mortality ratio—39 vs. 68%. A similar frequency of adverse events has been demonstrated, and AKI (acute kidney injury) and bone marrow aplasia were the most common AEs (adverse events). According to the study, adjuvant CFP-SBT therapy has a significant impact on survival among patients with VABP with a CRAB etiology [47].

The efficacy of TGC monotherapy or in combination with CFP-SBT in patients (N = 114) with a diagnosis of lower respiratory tract infection (LRTI) with the etiology of MDR *A. baumannii* was assessed in the Qin et al. study. Patients were randomly divided into the following two groups: those who received CFP-SBT monotherapy and those treated with a combination therapy (both groups equally, N = 57). The end-point of the study was a decrease in the concentration of inflammatory factors in the serum on the 14th day after the start of therapy (PCT—procalcitonin; CRP—C-reactive protein; IL-6—interleukine-6; and TNF-α—tumor necrosis factor α) and evaluation on the APACHE II scale (Acute Physiology and Chronic Health Evaluation II). The combination of TGC with CFP-SBT has been shown to reduce systemic inflammation (PCT, CRP, IL-6, and TNF-α had lower values in this group of subjects). Similarly, improvements in patients were reported on the APACHE II scale. The results clearly indicate the advantage of combination therapy over CFP-SBT monotherapy in the treatment of MDR *A. baumannii* etiological VABP [48].

A summary of clinical efficiency of cefoperazone-sulbactam treatment among patients with VABP is presented in Table 3.

## 3. Characteristics of Ceftolozane-Tazobactam

CEF-TAZ is a novel combination of the β-lactam antibiotic-ceftolozane and the β-lactamase inhibitor-tazobactam. It presents a promising effect in the treatment of MDR bacteria of the *Enterobacteriaceae* genus, including ESβL strains [49]. Furthermore, CEF-TAZ is also highly efficient against MDR *P. aeruginosa* and become an important option of treatment for infection of this etiology and may be used empirically [50]. According to the EMA (European Medicines Agency) guidelines, the indications for treatment using the combination are complicated urinary tract infections (cUTIs), including acute pyelonephritis (AP) and cystitis; complicated intra-abdominal infections (cIAIs); lung infections associated with mechanical ventilation (VABP); and HAP. Similar indications are presented by the FDA (Food and Drug Administration), taking into account mainly cUTIs and cIAIs. The drug should be administered every 8 h for 1 h lasting i.v. infusion at a dose of 1.5 g (in the ratio of 1 g of CEF and 0.5 g of tazobactam). The duration of treatment for cUTIs should be 7 days and for cIAIs 4–14 days. CEF-TAZ is capable of overcoming the efflux pump’s resistance mechanism and also porins mutations, which are increasingly frequent among MDR bacteria. However, it is inactivated by some β-lactamase enzymes such as carbapenemases [51]. Lung penetration by CEF-TAZ was assessed compared to PIP-TAZ. The concentration in epithelial lining fluid (ELF) was more than 8 mg/L, indicating that the growth in the microorganism in this case, *P. aeruginosa*, should be inhibited [52]. In the Sheffield et al. study, a lack of serious adverse events were reported among patients treated with CEF-TAZ, except for one patient, who experienced a worsening of a gout attack. The other most common side effects were headaches and digestive system problems, e.g., nausea and vomiting [51,53].


**Ceftolozane**


Ceftolozane (CEF) is a new β-lactam antibiotic from the group of V^th^-generation cephalosporins, and its name, according to IUPAC, is (6R,7R)-3-[[3-amino-4-(2- aminoethylcarbamoylamino)-2-methylpyrazol-1-ium-1-yl]methyl]-7-[[(2Z)-2-(5-amino-1,2,4-thiadiazol-3-yl)-2-(2-carboxypropane-2-yloxyimino)acetyl]amino]-8-oxo-5-thia-1-azabicyclo[4.2.0]oct-2-ene-2-carboxylate (PubChem). This preparation showed a promising activity profile against MDR strains and the relatively rare presence of adverse events [54]. It is highly active against Gram-negative bacteria (GNB) and presented an 8–16 times stronger activity against MDR strains than, for example, CPM or CAZ. CEF is also characterized by greater affinity to PBPs: 1b; 1c 2 and 3. The preparation was also shown to be superior to CAZ’s eradication activity against numerous bacterial strains [55]. The structural model of ceftolozane is shown in Figure 2A.


**Tazobactam**


Tazobactam (TAZ) is a β-lactamase inhibitor with the following IUPAC name: (2S,3S,5R)-3-methyl-4,4,7-trioxo-3-(triazol-1-ylmethyl)-4lambda6-thia-1-azabicyclo[3.2.0]heptane-2-carboxylic acid (PubChem). Molecule of TAZ is characterized by presence of a β-lactam ring in its structure. Tazobactam is a β-lactamase inhibitor that acts on the active site of the enzyme by irreversibly binding to it [52]. The combination of CEF with TAZ is highly effective against MDR microorganisms producing Ambler A class β-lactamases like ESβL, e.g., CTX-M-14 and CTX-M-15 [52]. However, it is inactive against bacteria with the expression of KPC and Ambler B class enzymes such as metallo-β-lactamases [19]. Research by Haidar et al. reported that among a group of 21 patients with MDR *P. aeruginosa* infections treated with CEF-TAZ, 71% achieved clinical cure. However, a worrying fact is that in three cases, the development of a resistance mechanism to the preparation was observed, and an increased expression of AmpC was also detected [56]. The structural model of tazobactam is shown in Figure 2B.

### 3.1. Clinical Efficiency of Ceftolozane-Tazobactam

#### 3.1.1. In Vitro Studies

Perez and coworkers presented findings of a study conducted as part of the MagicBullet trial, which was performed to assess the antibiotic resistance and efficacy of treatment for infections caused by XDR/MDR *P. aeruginosa* strains that occurred in several European countries, such as Greece, Italy, and Spain. In 2012–2015, a total of 121 *P. aeruginosa* isolates were collected, the highest numbers of PDR, MDR, and XDR isolates were identified in Greece, and the isolates were characterized by 22.6% resistance to CEF-TAZ and 24.5% resistance to CAZ-AVI. Resistant isolates expressed the VIM-2 MBL gene. In general, it was shown that the most active anti-*Pseudomonas* drugs in the study were COL (94.3% activity), CEF-TAZ (77.4%), and CAZ-AVI (75.5%) [57]. Sader et al. assessed the in vitro effectiveness of murepavadine against clinical isolates of XDR *P. aeruginosa* from 21 European countries as part of the SENTRY Antimicrobial Surveillance Program 2016–2017. Bacteria that were isolated from patients, including HAP/VABP, accounted for 63% of all isolates. In the general population of bacterial isolates, CEF-TAZ tested as a comparator showed an average MIC_50_ and MIC_90_ at the levels of 2 and >32 mg/mL, respectively, presenting the activity against 70.6% of isolates according to EUCAST (European Committee on Antimicrobial Susceptibility Testing) (COL MIC_50_ = 1 mg/mL; MIC_90_ = 2 mg/mL; activity = 93.6%). In the group of isolates from North America (N = 432), CEF-TAZ showed a higher activity of 86.8%, with MIC_50_ and MIC_90_ values of 1 and 8 mg/mL, respectively [58].

Carvalhaes et al. examined the in vitro susceptibility of, among others, VABP-derived clinical isolates of *P. aeruginosa* (N = 1531) and other *Enterobacteriaceae* (N = 2373) on CEF-TAZ using the microdilution method. The preparation exhibited an MIC_50_ and MIC_90_ in relation to the general population of *P. aeruginosa* at the levels of 0.5 and 2 mg/L and in the cases of the MDR and XDR strains with MIC_50_/MIC_90_ values of 1/8 and 2/16 mg/L. In turn, the activity against non-CRE (noncarbapenem-resistant *Enterobacterales*) *E. coli* and *K. pneumoniae* was as follows: 0.5/2 mg/L and 1/8 mg/L. According to EUCAST/CLSI (Clinical and Laboratory Standards Institute) guidelines, CEF-TAZ showed activity against 97.5% of *P. aeruginosa* isolates, making it the second most active after COL at 99.9%. The activities of CEF-TAZ against MDR and XDR *Pseudomonas* strains were similar at 87.9% and 82.9%. The activities against non-CRE *E. coli* and *K. pneumoniae* isolates oscillated around 80 and 90%, respectively [19]. This is reflected in a study conducted by Pfaller et al., which showed that CEF-TAZ is active against ESβL non-CRE *Enterobacterales* isolates between 81.4% and 89.4% (MIC_50_/MIC_90_ in the range of 0.5–8 mg/L). However, there was no significant effect on CRE isolates [59]. It is important is that among all tested BL and BL + BLI, CEF-TAZ showed the highest activity against MDR/XDR *P. aeruginosa* isolated from patients with VABP. This is a significant premise in the era of increasing antibiotic resistance among pathogens in the ESKAPE group [19].

The Idowu et al. study evaluated the in vitro efficacy of CEF-TAZ alone and in combination with a TOB homodimer against MDR and XDR *P. aeruginosa* strains. The study proved that the combination results in an increase in CEF-TAZ activity against *Pseudomonas* isolates and also prevents the development of resistance to the preparation, surpassing CAZ-AVI in this aspect [60].

The Karlowsky et al. study evaluated the susceptibility of bacterial isolates, including MDR *P. aeruginosa*, *K. pneumoniae*, and *E. coli*, from patients with LRTI (SMART—Study for Monitoring Antimicrobial Resistance Trends, conducted in the USA in 2018–2019). A total of 1237 *P. aeruginosa* isolates were tested. A high activity of CEF-TAZ against the pathogens was demonstrated (96%), which exceeded all other tested antibiotics, except AKC (96%). There was also a significant activity against *E. coli* and *K. pneumoniae* isolates (97.2% and 92.6%, respectively). Importantly, the preparation in these cases showed a superior activity profile than typical anti-*Pseudomonas* preparations, e.g., PIP-TAZ and CAZ. The medicament was active at the level of 81.6% and 86.8% against *P. aeruginosa* strains resistant to CAZ and PIP-TAZ, respectively. For ESβL *K. pneumoniae* and *E. coli* strains, the preparation was less active (67.1% and 86.5%). In turn, CEF-TAZ showed the highest activity against the following MDR strains: *E. coli* at 82.2% and *P. aeruginosa* at 76.2%, among other tested comparators. It was also observed that CEF-TAZ by its activity (96%) is equal to AKC and significantly exceeds PIP-TAZ (72.3%), CPM (78%), CAZ (76.6%), and even MER (72.9%) against *P. aeruginosa* isolates derived from ICU patients. In general, the treatment was equally effective when compared to AKC, which, when considering the risk of aminoglycoside nephrotoxicity, supports the CEF-TAZ treatment option [61]. Similar conclusions can be drawn from the Shortridge et al. study, which demonstrated that the in vitro activity against 1345 *P. aeruginosa* isolates derived from HAP/VABP patients was the highest in the monotherapies based on COL (>99%), AKC (98.1%), and CEF-TAZ (96.5%) [62]. These results correlate with another evaluation of *P. aeruginosa*’s sensitivity to CEF-TAZ and comparators [63]. Furthermore, CEF-TAZ monotherapy has been shown to be superior to CPM, CAZ, MER, and PIP-TAZ alone. Moreover, these BLs presented as comparable to the CEF-TAZ activity only in combination with AKC/COL [62].

The SMART study also assessed the susceptibility of *Enterobacteriaceae* (720 *P. aeruginosa* bacterial isolates, 338 *K. pneumoniae* isolates, and 291 *E. coli* isolates derived from ICU patients with confirmed LRTI (including VABP)) to, among others, CEF-TAZ. The study showed that *P. aeruginosa* isolates collected from ICU ward patients were characterized by the highest sensitivity to CEF-TAZ among the tested comparators, with activity inferior only to AKC (94% vs. 96%). In addition, relatively rare resistance to the preparation was noted in contrast to other BL and BL + BLI comparators (CEF-TAZ = 6%; CAZ = 27.4%; and PIP-TAZ = 31.7% resistance rates). It is worth emphasizing that CEF-TAZ showed significant activity against strains resistant to, among others, MER (83.1%), PIP-TAZ (82.5%), or CAZ (78.2%). It is also worth noting, that the preparation had relatively high activity against MDR *P. aeruginosa* (71.4% of strains from ICU) and pan-β-lactam-non-susceptible *P. aeruginosa* (resistant to III^rd^- and IV^th^-generation cephalosporins, ATM, carbapenems, and PIP-TAZ), where it was effective against 65.6% of isolates (collected from ICU patients) [64].

The Candel et al. study evaluated the in vitro susceptibility of bacterial pathogens, including isolates from patients suffering from VABP to cefiderocol and comparators, including CEF-TAZ. A total of 20,911 isolates were collected, of which 34.4% were from VABP infections. *Enterobacteriaceae* accounted for 51.2% of all isolates, the most numerous species were *K. pneumoniae* and *E. coli*. Nonfermenting pathogens constituted 48.8% of isolates, the most numerous were *P. aeruginosa*, *A. baumannii*, and *S. maltophila*. There was no activity against *A. baumannii* and *S. maltophila* from the CEF-TAZ, while the preparation showed significant activity in relation to isolates sensitive to the carbapenems *E. coli*, *K. pneumoniae*, and *P. aeruginosa* (97%, 86.9%, and 97.5%, respectively). Among the CRE isolates, a lower activity was observed. A moderate susceptibility rate was observed only in the case of *P. aeruginosa* (CRPA), at the level of 48.1%. The study proved the increasing resistance to CEF-TAZ among ESKAPE pathogens, while cefiderocol is a promising response to this trend (with a high activity against carbapenem-resistant *S. maltophila* and *P. aeruginosa* of the order of >90%). This well reflects the MIC_50_/MIC_90_ values for the pathogens mentioned, respectively, for cefiderocol vs. CEF-TAZ: *P. aeruginosa*—0.25/0.5 mg/L vs. 0.5/8 mg/L and *K. pneumoniae*—0.25/2 mg/L vs. 0.5/> 64 mg/L [65]. A summary of the in vitro antimicrobial activity of ceftolozane-tazobactam is presented in Table 4.

#### 3.1.2. Clinical Trials

The effectiveness and safety of the treatment for HAP/VABP with CEF-TAZ vs. MER were assessed in a phase 3 randomized, double-blind, and controlled trial (ASPECT-NP). The primary end-point of the study was, among others, 28-day mortality for any cause. The secondary end-point was the clinical response at TOC (test-of-cure: 7–14 days after EOT (end of treatment)) in the microbiological ITT population (mITT—microbiological intention to treat population). The patients were randomized into two groups, as follows: treated with CEF-TAZ (N = 362) and treated with MER (N = 364). Those in the first group received 3 g of CEF-TAZ (2 g of CEF and 1 g of TAZ) and the second 1 g of MER as 1-h i.v. infusions every 8 h for 8–14 days. In the general population, the predominant form of the disease was VABP (71%). The dominant isolated pathogens were represented by *K. pneumoniae*, *E. coli*, and *P. aeruginosa* (including strains of ESβL). The 28-day mortality rates were as follows: 24% for CEF-TAZ vs. 25.3% for MER treatment. In turn, the obtained clinical responses at TOC, respectively, were 54% vs. 53%. Another relevant end-point was microbial eradication assessed in the mITT population at TOC (73.1% vs. 68%, respectively). In both cases, a similar eradication activity of the preparations was observed against MDR *P. aeruginosa* (~54%), XDR *P. aeruginosa* (40%), and against ESβL *Enterobacteriaceae* (57.1% vs. 61.6%). The adverse events profile (TEAE) was also similar in both groups at 11% vs. 8% (of which severe TEAE accounted for 8/38 and 2/27 of the total amount of TEAE, respectively). According to the study, CEF-TAZ has a comparable antimicrobial profile to MER in VABP therapy, indicating a potential for its application in an empirical therapy [66,67,68].

Pogue et al. revealed the effectiveness of CEF-TAZ therapy in relation to the comparators, as follows: aminoglycosides (gentamicin [GEN], tobramycin [TOB], and AKC) and COL in the treatment of MDR/XDR *P. aeruginosa*. In this multicenter, retrospective, and observational cohort study, patients were randomly divided (N = 200) into two (I and II) groups (both equally, N = 100). The dominant type of infection was VABP (63% of patients were mechanically ventilated, and VABP developed in 52%). The need to switch to combination therapy (adding antibiotic—most often MER, β-lactams, and PIP-TAZ in I or COL in II) occurred significantly more frequently in the cohort receiving aminoglycoside or polymyxin than CEF-TAZ (75% vs. 15%). Importantly, better clinical cure values were achieved in the CEF-TAZ cohort (81% vs. 61%), as was the issue of hospital mortality (20% vs. 25%). The difference was also related to the safety profile of the therapy, especially nephrotoxicity understood as AKI (6% vs. 34%). Nearly 7% of patients treated with aminoglycosides/polymyxins required renal replacement therapy. The study highlighted the benefits of CEF-TAZ in the treatment of VABP. The CEF-TAZ therapy was found to have lower kidney toxicity than the comparator, as well as superior effectiveness against MDR/XDR *P. aeruginosa* [69].

Mogyorodi et al. compared the efficacy of CEF-TAZ or COL therapy in the treatment of VABP with the etiology of XDR *P. aeruginosa* among patients hospitalized in an ICU ward. A total of 51 patients were enrolled in the trial and were randomly assigned to the following two groups: those receiving CEF-TAZ (N = 18) and those receiving COL (N = 33). The end-points of the study were clinical cure (understood as the resolution of symptoms and signs of infection), microbiological cure (eradication or persistent colonization), and 28-day mortality ratio and frequency of AEs. The median duration of treatment was 7 days in the CEF-TAZ group and 9 days in the COL group. The need for combination therapy was observed in 32/33 (97%) cases in the COL group (mainly the addition of inhaled COL) and in 8/18 (44%) cases in the CEF-TAZ group. Clinical success was achieved to a greater extent in the CEF-TAZ group than in the COL group (72.2% vs. 30.3%). Microbiological eradication occurred in 44.4% of patients in the first group and in only 15.2% of patients treated with COL. There were no significant differences in the 28-day mortality rate (27.8% vs. 33.3%). The AEs were significantly more frequent in the COL population (72.7%) vs. CEF-TAZ (55.5%). The development of AKI was more frequent in patients treated with COL than CEF-TAZ (48.5% vs. 11.1%) [70]. In a retrospective clinical study conducted in 2016–2018 in 22 Italian hospitals, the therapeutic effectiveness of CEF-TAZ in the treatment of, among others, nosocomial infections with the etiology of *P. aeruginosa* was assessed. A total of 101 patients were enrolled in the study (VABP cases accounted for 12, in turn HAP = 20). Overall, 75% of patients presenting symptoms of LRTI achieved clinical success, 8 patients from the VABP group developed clinical failure, which accounted for 8/17 of all cases of failure. Side effects were reported in only 3/101 patients. Particularly important is that it was shown that out of the tested bacterial isolates, as much as 50.5% were classified as XDR *P. aeruginosa* (among others, high resistance to fosfomycin (FOS = 89.1%), MER (77.2%), PIZ-TAZ (77.2%), and GEN and TOB (50.5%)) [71].

A study in the Canadian Leadership on Antimicrobial Real-Life use (CLEAR) registry conducted in Canada, in 2019–2020, evaluated the clinical effectiveness of CEF-TAZ in the treatment of, among others, patients suffering from HAP and VABP and hospitalized on and off in the ICU. The study included 51 patients, of which 8 were diagnosed with VABP and 19 were treated for HAP. Based on the in vitro isolates tested, the average activity of the preparation was determined to be 88.2%. Patients treated for VABP received CEF-TAZ due to the resistance of bacteria to previous therapy (8/8), all of them had microbiologically confirmed *P. aeruginosa* infection. In three cases, combination therapy with fluoroquinolone and aminoglycoside was implemented. There were three deaths; in addition, one of the patients developed clinical failure in the course of therapy. Clinical success (improvement and clinical cure) was observed in the remaining patients. The duration of treatment was usually in the range of 7–10 days. Adverse events were reported in 1/8 cases, and the development of neutropenia occurred [72].

In another retrospective multicenter clinical trial, Gallagher et al. investigated the efficacy of the treatment of MDR *P. aeruginosa* infections with CEF-TAZ. The primary end-points of the study were 30-day mortality and the in-hospital mortality rate. For secondary points, the following were chosen: clinical cure (resolution of the signs and symptoms of infection) and microbiological eradication (negative culture in EOT). A total of 205 patients were enrolled, of which 121 were suffering from pneumonia (with VABP cases accounting for 63). *P. aeruginosa* isolates were shown to have a high rate of antibiotic resistance (%) to carbapenems (96.8%), PIP-TAZ (94.2%), ATM (92.8%), and CPM/CAZ (85.6%). In the general population, the total mortality was 19%, the VABP population achieved clinical success in 50%, microbiological eradication was confirmed in 53.4% of patients, 22 out of 39 deaths related to patients with VABP [73].

In the multicenter retrospective study by Holger et al., the efficacy of CEF-TAZ therapy vs. BAT (including the use of PIP-TAZ, CPM, MER, CAZ/AVI, CAZ, or COL) was evaluated in the treatment of LRTI with the etiology of MDR/XDR *P. aeruginosa*. The study enrolled 206 patients, who were then randomly assigned into two groups: those who received CEF-TAZ (N = 118) and those treated with BAT (N = 88). The percentage of VABP in both groups was 52.5% and 38.6%, respectively. In the general populations of CEF-TAZ and BAT, the primary end-points were as follows: clinical failure—23.7% vs. 48.9%; 30-day mortality rate—15.3% vs. 20.5%; and TEAE frequency—10.2% vs. 33%. Taking into account these results and a much higher percentage of VABP in the CEF-TAZ treatment group, it can be concluded that the preparation showed a significant advantage over BAT in the treatment of MDR/XDR *P. aeruginosa* etiology LRTI. The therapy was also characterized by greater safety and a lower percentage of TEAEs [74].

A summary of the clinical efficiency of ceftolozane-tazobactam therapy among patients with VABP is presented in Table 5.

## 4. Characteristics of Ceftazidime-Avibactam

Ceftazidime (CAZ) is a III^rd^-generation cephalosporin which, in combination with the non-β-lactam β-lactamase inhibitor avibactam (AVI), is active against GNB, especially *Enterobacteriaceae* [21]. Worse activity is observed against nonfermenting *P. aeruginosa* due to its resistance of 2.9–18%, as well as against *A. baumannii*, which has a resistance to CAZ-AVI that reaches 50% [75]. However, AVI, in combination with antibiotics, decreases the MIC value for *P. aeruginosa* by nearly four times. The effect of CAZ-AVI has been demonstrated also against anaerobic bacteria, e.g., *Prevotella* spp., *Bacterioides fragilis*, *Clostridium perfringens*, and *Porphyromonas* spp. [76]. The combination of CAZ-AVI with other antibiotics achieved promising results in the terms of the eradication of *K. pneumoniae* strains colonizing intestines. Success was achieved in 11/12 cases despite the presence of a KPC resistance mechanism among pathogens [77]. CAZ-AVI, according to EMA indications, can be used in the following cases: cIAIs, cUTIs, including AP; nosocomial pneumonia (HAP); and VABP, as well as in the case of bacteremia associated with the above disease states. FDA guidelines also indicate the use of this combination in the conditions mentioned. Moreover, CAZ-AVI can also be used in pediatric patients over 3 months of age in treatment of cUTIs and cIAIs. The drug is administered at a ratio of 2 g of CAZ to 0.5 g of AVI in i.v. infusions lasting 2 h every 8 h in adult patients with normal renal function (CrCl > 50 mL/min and eGFR > 50 mL/min/1.73 m^2^). For pediatric patients, the dosage is altered, as follows: children between 3 and 6 months of age should receive 40 mg/kg of CAZ and 10 mg/kg of AVI. However, above 6 months of age, 50 mg/kg of CAZ and 12.5 mg/kg of AVI should be administered. The most common adverse events revealed in clinical trials included gastrointestinal problems, such as abdominal pain, diarrhea, and constipation, as well as headaches, injection site reactions, fever, and increased levels of transaminases. Renal failure and diarrhea have been reported rarely [78,79]. In Drwiega et al.’s study, the effects of different ways of administering CAZ-AVI were assessed. Patients received a bolus of the drug at a dose of 20 mg/kg body weight, and then after 60 min a continuous infusion at a dose of 60 mg/kg body weight per day. Pulmonary penetration of CAZ in studies among patients hospitalized due to VABP achieved better values during continuous infusion in contrast to bolus infusion. The median ELF concentration was 12 µg/mL, and the ELF-to-plasma penetration ratio was 0.42 [80].


**Ceftazidime**


Ceftazidime (CAZ) is a β-lactam antibiotic belonging to the group of III^rd^-generation cephalosporins with the IUPAC name (6R,7R)-7-[[(2Z)-2-(2-amino-1,3-thiazol-4-yl)-2-(2- carboxypropane-2-yloxyimino)acetyl]amino]-8-oxo-3-(pyridin-1-ium-1-ylmethyl)-5-thia -1-azabicyclo[4.2.0]oct-2-ene-2-carboxylate (PubChem). Similarly to other cephalosporines, the CAZ molecule binds to PBP proteins preventing the cross-linking of peptidoglycan in the microorganism wall, resulting in cell death [21]. The structural model of ceftazidime is shown in Figure 3A.


**Avibactam**


Avibactam (AVI) is a non-β-lactam β-lactamase inhibitor, and its IUPAC name is [(2S,5R)-2-carbamoyl-7-oxo-1,6-diazabicyclo[3.2.1]acetan-6-yl] hydrogen sulfate (PubChem). The inhibitor molecule has the ability to reversibly covalently attach to the active site of serine β-lactamases [52,81]. It is active against *Enterobacteriaceae* producing: ESβL, KPC, AmpC, and OXA-48. However, it is ineffective against metallo-β-lactamases (group B, according to Ambler, e.g., VIM, IMP, and NDM). Therefore, it is active against Ambler classes A, C, and some enzymes from group D producers [21,78]. The resistance to CAZ-AVI is connected with structural point mutations in β-lactamase molecules, as well as changes in the membrane proteins and efflux pump expression [75]. The structural model of avibactam is shown in Figure 3B.

### 4.1. Clinical Efficiency of Ceftazidime-Avibactam

#### 4.1.1. In Vitro Studies

The Sader et al. study evaluated the activity of CAZ-AVI against Gram (−) bacterial isolates collected from patients (ICU and other hospital wards) presenting, among others, symptoms of VABP. In total, 18,864 isolates were tested, of which 435 derived from VABP infections. The results obtained by CAZ-AVI were very promising. The growth of 99.9% of the *Enterobacteriaceae* was inhibited at MIC levels lower than 8 mg/L. In turn, the MIC_50_/MIC_90_ determined for isolates derived from VABP were 0.12/0.5 mg/L. The KPC isolates and carbapenem-resistant *Enterobacteriaceae* also had high susceptibility to CAZ-AVI at MIC_50_/MIC_90_ = 0.5/2.0 mg/L. The CAZ-AVI activity against *P. aeruginosa* isolates collected from VABP was also significant at MIC_50_/MIC_90_ = 2.0/4.0 mg/L, respectively. However, individuals resistant to CAZ or MER and MDR isolates were less susceptible (4.0/16.0 mg/L). XDR *P. aeruginosa* achieved higher MIC values similar to *A. baumannii* (8.0/32.0 and 16.0/>32 mg/L, respectively). Importantly, AVI significantly increased the sensitivity of *P. aeruginosa* to CAZ (especially with ICU-ward-derived isolates), where this pathogen was initially sensitive in ~78% and in combination with AVI it increased to 95.6% [82]. Perez et al. evaluated the susceptibility of clinical isolates of MDR and XDR *P. aeruginosa* (collected from patients suffered from VABP) to a number of antibiotics including CAZ-AVI. The preparation was characterized by an MIC_50_/MIC_90_ at the level of 2.0/16.0 mg/L (sensitivity of 75.5%), which makes it comparable to the CEF-TAZ described above (77.4%) [57].

Another study evaluated the CAZ-AVI activity separately and in combination with ATM against *K. pneumoniae* CRE, NDM, and KPC isolates in vitro, as well as in vivo in a murine model of infection. A total of 47 *Klebsiella* isolates were collected, of which 16 were KPC-2, 1 was OXA-232, 28 were identified as NDM, and 2 were identified as KPC-2 + NDM. CAZ-AVI was highly active in vitro against KPC-2 isolates (MIC = 0.4–0.8 mg/dL) and OXA (0.2 mg/dL). The preparation showed lower activity against NDM isolates (MIC in the range of 0.5 to 256 mg/L) and against KPC and NDM isolates (MIC value of 8 and 128 mg/L, respectively). Significantly, in most cases (90%) synergistic interactions between CAZ-AVI and ATM were demonstrated. The combination led to a decrease in the MICs of both compounds and restored ATM activity against KPC and NDM isolates [83].

The in vitro susceptibilities of *Enterobacteriaceae* clinical isolates (including *P. aeruginosa*) collected from a series of studies (e.g., REPROVE) were assessed for many antibiotics, including CAZ-AVI. The combination obtained MIC_50_/MIC_90_ values for individual pathogens causing VABP as follows: MDR *P. aeruginosa* (8/64 mg/L, strain sensitivity 34.8%) and MDR *K. pneumoniae* (0.5/1.0 mg/L, sensitivity 75%). The REPROVE study confirmed previous reports on the superiority to comparators of the antimicrobial activity profile of CAZ-AVI against MDR GNB causing VABP [84,85].

In the next study, the in vitro activity of CAZ-AVI was evaluated in relation to GNB bacterial isolates collected from patients hospitalized due to pneumonia (including VABP) in 2011–2015. A total of 11185 isolates were tested, of which 1097 were collected from patients suffering from VABP. Subsequently, the sensitivity profiles to the drugs CAZ-AVI and comparators were determined. CAZ-AVI showed high activity against *P. aeruginosa* (MIC_50_/MIC_90_ = 2/4 mg/L), including MER-resistant strains (4/16 mg/L) and PIP-TAZ-resistant MDR strains (4/16 mg/L in both strains). The XDR pathogens were characterized (similarly to Sader et al.’s 2015 study) by higher MICs (8/32 mg/L). This trend was observed also among *A. baumannii* isolates (16/>32 mg/L). Pathogenic CREs were highly susceptible to CAZ-AVI, with MIC_50_/MIC_90_ at 0.5/2 mg/L. Among ESβL strains, there was also high sensitivity to combinations, as follows: *K. pneumoniae* (0.25/1 mg/L) and *E. coli* (0.12/0.5 mg/L). Overall, a high percentage of CAZ-AVI susceptibility was demonstrated for most GNB isolates derived from VABP, e.g., *P. aeruginosa* (97.8% including MDR/XDR = 87.5%), ESβL *K. pneumoniae* (93.1%), and ESβL *E. coli* (100%). The study showed noninferior activity of the preparation to comparators against *Enterobacteriaceae* and significantly weaker in relation to *A. baumannii* (e.g., COL = 1/2 mg/L) [86].

The Candel et al. study evaluated the in vitro activity of cefiderocol and comparators, including CAZ-AVI against GNB isolates obtained in the years 2013–2018 as part of the SIDERO-WT and SIDERO-Proteeae studies. A total of 20,911 isolates were collected, of which 34% were collected from patients with VABP (mainly *K*. *pneumoniae*, *P*. *aeruginosa*, and *A*. *baumannii)*. The activity of CAZ-AVI against isolates sensitive to carbapenems (derived mainly from VABP infections) was at the level of 98–99.6%. CRE pathogens were less sensitive to CAZ-AVI (*K. pneumoniae* at 70.5% and *P. aeruginosa* at 46.1%). Cefiderocol or COL presented activity profiles against *A. baumannii* and *S. maltophila* superior to CAZ-AVI [65]. The study showed some limitations of CAZ-AVI activity in relation to *A. baumannii* or *S. maltophila*, which were also reported by other researchers [82,86]. A summary of the in vitro antimicrobial activity of ceftazidime-avibactam is presented in Table 6.

#### 4.1.2. Clinical Trials

In a multicenter, randomized, double-blind, and phase 3 study—REPROVE (Randomized Trial of ceftazidime-avibactam Versus Meropenem for Treatment of Hospital-Acquired and Ventilator-Associated Bacterial Pneumonia)—Torres et al. evaluated the CAZ-AVI therapeutic efficiency vs. MER for the treatment of nosocomial infections (including VABP). Patients were randomly divided into the following two groups: those who received CAZ-AVI (N = 356, CAZ-AVI 2000 + 500 mg every 8 h) and those treated with MER (N = 370, MER 1000 mg every 8 h). The percentage of VABP infections per group were 33 vs. 35%. The predominant pathogens causing infections were identified as: *K. pneumoniae* and *P. aeruginosa* (total ~70% of isolates). Clinical cure (the primary end-point of the trial) was achieved in 68.8% (CAZ-AVI) vs. 73% (MER) of the mITT population (clinical modified intention-to-treat—patients meeting entry trial criteria with confirmed GNB etiology infection), in the clinically evaluable population (CEP: mITT patients after receiving treatment), the results for clinical cure were as follows: 77.4% vs. 78.15%. Patients with confirmed VABP achieved clinical success in the mITT and CEP populations for the CAZ-AVI and MER treatments, respectively, at 70.3% vs. 74.2% and 77.5% vs. 75.9%. Both preparations were characterized by similar activities against pathogens isolated from patients, both Gram (−) and (+). Similarly, the 28-day mortality ratio was 9 vs. 7%, and the frequency of AEs was 75% vs. 74% (the most common were diarrhea, hypokalemia, and anemia). According to the results of the study, CAZ-AVI showed noninferior activity to MER, which is a prerequisite for its use in carbapenem-saving therapy for VABP and HAP [87,88].

Interesting insights also come from the Shi et al. study, which assessed the effectiveness of CAZ-AVI vs. TGC therapy in patients hospitalized in an ICU ward due to HAP/VABP with the etiology of CRE *K. pneumoniae*. The study included 105 patients (% of confirmed VABP = 71.4) who were randomized into two groups: those obtaining TGC (N = 62, the scheme included 200 mg in loading dose and then a maintenance dose) and treated CAZ-AVI (N = 43, regimen of 2000 + 500 mg every 8 h). Clinical success was defined as the resolution of clinical symptoms in combination with the normalization of nonmicrobiological components, for example, laboratory parameters. Microbiological success was defined as the eradication/negative cultures at the end of the patient’s hospitalization. CAZ-AVI therapy has been shown to be associated with a higher percentage of clinical cure rates, as follow: 51.2% vs. 29%, respectively. Similarly, the issue of microbiological success was presented, as follows: 74.4% vs. 33.9%. There was no significant difference in 28-day survival (69.8% vs. 66.1%). Importantly, CAZ-AVI therapy was also associated with a lower percentage of AEs (mainly diarrhea) than TGC treatment, as follows: 7% vs. 27.4%, respectively. The study proved that CAZ-AVI could be an alternative treatment option for critically ill ICU ward patients with VABP of CRE pathogens etiology [89].

A retrospective observational cohort study conducted in Greece in a population of critically ill ICU patients (diagnosed with VABP) demonstrated the superiority of CAZ-AVI therapy over other treatment regimens for infections with a CRE etiology. Patients were randomly assigned into the following two groups: those receiving CAZ-AVI (N = 41) and those receiving BAT (N = 36). A total of 26 cases of VABP were reported (19 vs. 7 among populations). CAZ-AVI presented better results in the field of clinical cure among patients (80.5% vs. 52.8% for BAT). Moreover, it also achieved a microbiological eradication ratio superior to its comparator (94.3% vs. 67.7%).

The 28-day survival rate also favored CAZ-AVI therapy (85.4% vs. 61.1%), in turn, there was no difference in the frequency of AEs. CAZ-AVI was shown to have a higher efficacy rate than COL-based regimens, which is crucial because of colistin’s nephrotoxicity and limited therapeutic use in patients with impaired renal function [90]. These conclusions are supported by van Duin et al.’s study, in which the superiority of CAZ-AVI therapy over colistin was demonstrated in the field of in-hospital mortality (CAZ-AVI treated patients—9%; COL treated patients—32%) in the therapy of CRE etiology infections [91].

Zheng et al., in turn, investigated the efficacy of CAZ-AVI monotherapy and CAZ-AVI-based combination therapy in the treatment of CRE etiology infections. Patients (N = 62) were randomly divided into the following two groups: those treated with CAZ-AVI alone (N = 21) and those receiving combination therapy (N = 41). The primary end-point of the study was the 30-day mortality of patients from any cause. Mortality in the first population was 47.6% vs. 24.4% in the combination group. A similar advantage of polytherapy was demonstrated in the evaluation of microbiological eradication ratio: 42.9% vs. 61%. Particularly promising clinical parameters (related to the reduction in 30-day mortality) were obtained by schemes based on the combination of CAZ-AVI with carbapenems (MER and IMI), TGC, FOS, and ATM [92]. There are also reports on the potential effectiveness of combined therapies that consisted of CAZ-AVI + ATM or CAZ-AVI + MER + ertapenem (double-carbapenem therapy) in the treatment of VABP with the etiology of PDR *K. pneumoniae*. The therapeutic regimens used led to the alleviation of parameters on both the SOFA (Sequential Organ Failure Assessment) and CPIS (Clinical Pulmonary Infection Scale) scales. This is an important premise in the topic of using CAZ-AVI + ATM in VABP empirical therapy as a beneficial option to increase the chances of survival for patients [93].

The Burastero et al. reported the potential use of CAZ-AVI plus FOS/AKC/trimethoprim-sulfamethoxazole (TRM-STX)/MER therapy regimen in the treatment of VABP with an etiology of *P. aeruginosa* DTT (DTT—difficult-to-treat, strains nonsusceptible to β-lactams and fluoroquinolones), *Burkholderia cepacia*, and *S. maltophila* complicated by SARS-CoV2 infection. It was reported that in critically ill patients with mentioned viral coinfection, such a treatment scheme led to significant microbiological eradication (achieved in 14/23 of the cases studied). The problem remaining is a high mortality ratio, recorded at the level above 60% despite ICU hospitalization. Because of the challenges of treating infections of this etiology, the completion of the microbiological eradication in 3/6 cases of *S. maltophila* infection is a significant premise for the continuation of research in this field [94].

The effectiveness of CAZ-AVI therapy in the treatment of infections with the etiology of MDR/XDR *P. aeruginosa* was also assessed in a study by Corbella et al. The study population (N = 61) included patients presenting, among others, LRTI (34.4%). It was confirmed that predominant etiological factor of VABP infections was *P. aeruginosa* (91.8% of isolates were identified as MDR and 8.2% as XDR). The median duration of therapy was 7 days, and in 47% of cases an additional drug was administered, including MER, COL, and ATM. Clinical cure was achieved on day 14 of therapy in 54.1% of cases, while the 30-day mortality rate was 13.1%, recurrence by the day 90 occurred in 12.5% of cases. Promising results were also obtained when CAZ-AVI was combined with ATM (total absence of recurrent infections) and in combination with AKC (100% 30-day survival ratio vs. 93.8% in case of CAZ-AVI monotherapy). Interestingly, the combination of CAZ-AVI with COL did not achieve better parameters than CAZ-AVI monotherapy [95].

Another multicenter, retrospective, and cohort clinical trial evaluated the therapeutic efficacy of CAZ-AVI in the treatment of carbapenem-resistant *Enterobacteriaceae* and *P. aeruginosa* etiology infections. The study achieved similar results to Corbella et al., as follows: 30-day mortality ratio of 17.2%, recurrence by the day 30 ratio of 5.9%, and clinical success reached in 70.9% of patients (N = 203). Particularly important is that these results were similar among patients in the general population, those infected with *P. aeruginosa*, and among patients with CRE infection. The LRTI accounted for 37.4% of all infections, and *K. pneumoniae* accounted for as much as 43.8% of all isolates (including 63.2% of all CRE isolates). The MIC_50_/MIC_90_ values of CAZ-AVI were *K. pneumoniae* CRE 2/4 mg/L and 2/6 mg/L for *P. aeruginosa*. Both studies demonstrate the high efficacy of CAZ-AVI monotherapy in the treatment of GNB MDR infections, including mechanically ventilated patients. This is an important premise in the era of increasing resistance to carbapenems and anti-*Pseudomonas* preparations [96].

A summary of the clinical efficiency of ceftazidime-avibactam therapy among patients with VABP is presented in Table 7.

## 5. Characteristics of Cefiderocol

Cefiderocol (CFD) is a newly synthetized siderophore cephalosporin antibiotic (IUPAC name: (6R,7R)-7-[[(2Z)-2-(2-amino-1,3-thiazol-4-yl)-2-(2-carboxypropane- 2-yloxyimino)acetyl]amino]-3-[[1-[2-[(2-chloro-3,4-dihydroxybenzoyl)amino]ethyl] pyrrolidin-1-ium-1-yl]methyl]-8-oxo-5-thia-1-azabicyclo[4.2.0]oct-2-ene-2-carboxylate (PubChem)) (Figure 4). CFD presents the unique mechanism of action which uses bacteria’s ability to collect iron ions from the external environment and transport them into the cells. The siderophore antibiotic molecule after active transport through iron ions carriers into periplasmatic space attach to PBP proteins and then disrupt the structure of the bacterial wall, which ends in cell lysis. It was revealed that CFD can also be taken into bacterial cell via porines, which is the mechanism of penetration into the cell similar to that exhibited by classical β-lactam preparations [97]. Figure 5 shows the mechanism of action of cefiderocol in comparison to the β-lactam β-lactamase inhibitor combination against a bacterial cell on molecular level. The spectrum of action of CFD includes nonfermenting *P. aeruginosa*, *A. baumannii*, *S. maltophila*, and other *Enterobacteriaceae*, as well as MDR bacteria [98]. The MIC_90_ value for *Enterobacterales*, nonfermenting bacilli and *Proteaceae* obtained were in the range of 0.12 to 2 mg/L. Moreover, 95% to even 100% of the strains remained susceptible [22]. According to EMA, CFD is indicated for the treatment of infections caused by Gram (−) aerobic bacteria and should be used when previous therapeutic options have failed. The FDA recommendations describe that the drug can be used to treat HAP, VABP, and cUTIs including AP in patients over 18 years of age. Doses of 2 g should be administered every 8 h as an i.v. infusion lasting 3 h in patients with normal renal function (defined as CrCl in the range of 60–119 mL/min). Resistance to CFD on molecular level may be associated with the Ton-B-dependent iron transporter mutations and mutations in *ampC* genes. It was also observed that the MIC value for this preparation may increase when the microorganism retains NDM resistance (it acts only on 58% of *Enterobacterales* NDM isolates; considering that the MIC value ≤ 4 μg/mL indicates the susceptibility of the bacteria) [99]. CFD molecule is stable against hydrolysis by serine β-lactamases like KPC and also against MBL [100]. The penetration of CFD into the lungs has been also investigated. Patients suffering from VABP received CFD at a dose of 2 g every 8 h in a 3 h infusion, achieving ELF values of 7.63 mg/L immediately after the infusion and 10.40 mg/L for 2 h after the end of the drug’s administration. The ratio of ELF to the unbound drug concentration in plasma was 0.212 at the end of the 3 h of continuous infusion and 0.547 in 2 h after the end of the infusion. The above results demonstrate that the permeability of the drug is adequate for the treatment of pneumonia [101]. The most common AEs connected with the treatment include skin changes, such as rashes, headaches, abdominal pain, throat and mouth pain, fever, hypertension, nausea, vomiting, diarrhea, constipation, cough, and an increase in liver enzymes and creatine phosphokinase plasma activity and leukocytosis [102,103].

### 5.1. Clinical Efficiency of Cefiderocol

#### 5.1.1. In Vitro Studies

The in vitro activity of CFD on GNB isolates was assessed in the course of the SIDERO-WT study conducted in 2014–2017. A total of 19,119 isolates were assessed, among which *E. coli*, *K. pneumoniae*, *P. aeruginosa*, and *A. baumannii* were dominant. Overall, a promising CFD activity profile was demonstrated for the isolates tested; the MIC_90_ for *K. pneumoniae* over the years was assessed as 1 µg/mL, while those of *P. aeruginosa* at 0.5 μg/mL and *A. baumannii* in the range 1–4 μg/mL. It is worth emphasizing that the preparation was also highly active against *S. maltophila* (MIC_90_ = 0.25–0.5 µg/mL). The CFD was also shown to be highly active against MER-resistant isolates, as follows: *P. aeruginosa* (MIC_90_ = 0.5–1 µg/mL) and *A. baumannii* (MIC_90_ = 1–4 µg/mL). The preparation within 3 years of the study showed excellent activity against CRE isolates, as follows: *P. aeruginosa* (99.7–100%) and *A. baumannii* (91%-96.9%). The activity of CFD was superior to the evaluated comparators, such as CEF-TAZ and CAZ-AVI, or even COL (MIC_90_ > 8 µg/mL) against the tested MER-resistant *Enterobacteriaceae* isolates [20,104]. In the SIDERO-CR study, the CFD activity against carbapenem-nonsusceptible clinical isolates of *Enterobacteriaceae* and *P. aeruginosa*, *A. baumannii* and *S. maltophila* (the study included MDR isolates) was evaluated. The preparation showed the highest activity against *Enterobacteriaceae* (97% for CFD vs. 77% for CAZ-AVI vs. 77.8% for COL), similarly for nonfermenting bacilli: *P. aeruginosa* (99.2% CFD vs. 99.6% COL), *A. baumannii* (90.9% CFD vs. 94.6% COL). There was a 100% activity against *S. maltophila* isolates insensitive to carbapenems, which made CFD the most active of the tested compounds. The MIC_90_ for CFD was determined in study at the value of 4 µg/mL, however the compound inhibited the growth of 97% of isolates already at subinhibitory concentration [105].

Another in vitro study evaluated the activity of CFD and comparators against GNB isolates obtained from patients in the SIDERO-WT and SIDERO–Proteeae studies. Of the 20911 isolates evaluated, nearly 34.4% were collected from patients diagnosed with VABP. Among the isolates predominant were *Enterobacteriaceae*, especially *Klebsiella* spp., nonfermenting bacilli represented 48.8% of VABP isolates (mainly *A. baumannii* and *P. aeruginosa*). The CFD activity against VABP isolates was assessed. The percentages of susceptible bacterial strains to CFD are presented as follows: *E. coli*—99.6%; *Klebsiella* spp. [CR]—63.8%; carbapenem-sensitive [CS]—97.9%, *P. aeruginosa* (CR: 97.1%, CS: 99.6%), *A. baumannii* (CR: 91%, CS: 94.4%), *S. maltophila* (CR: 99.6%). CFD showed higher activity than comparators: CAZ-AVI, CEF-TAZ. Only COL showed activity against *A. baumannii* and *S. maltophila* outside CFD [65]. Another in vitro study evaluated the activity of CFD and comparators (BL + BLI) against GNB fermenting and nonfermenting isolates (including those derived from HAP/VABP positive patients) under the SENTRY Antimicrobial Surveillance Program for 2020. CFD was the most active of the tested preparations (including COL) against *P. aeruginosa* (MIC_50_/MIC_90_ = 0.12/0.5 mg/L) and including XDR and MER-VAB resistant isolates (0.12/1 mg/L for both isolates). In addition, it demonstrated, similar to COL, activity against isolates resistant to all tested BL + BLI connections (CEF-TAZ, CAZ-AVI, PIP-TAZ, and imipenem-relebactam). Moreover, it was also the most active formulation against *A. baumannii* (0.25/1 mg/L), including MER-resistant isolates (0.5/2 mg/L). Interestingly, CFD showed the same activity against *S. maltophila* isolates as TRM-STX (0.12/0.5 mg/L), which makes these two preparations the most active among the tested substances [106]. A summary of the in vitro antimicrobial activity of cefiderocol is presented in Table 8.

#### 5.1.2. Clinical Trials

A randomized, double-blind, and phase 3 study—APEKS-NP—evaluated the efficacy of i.v. CFD therapy (2000 mg every 8 h) vs. MER (2000 mg every 8 h) in the treatment of nosocomial infections (including HAP and VABP). Both preparations were administered in combination with linezolid. Patients (N = 300) were randomized into the following two groups: those treated with CFD and those treated with MER (N = 148 and N = 152, respectively). The percentages of VABP infections in both groups was 41% vs. 44%. Among these patients, the most common etiological factors of infection were *K. pneumoniae*, *P. aeruginosa*, and *A. baumannii*. Moreover, *A. baumannii* was shown to be predominant carbapenemase (70%) and EsβL producer (43–67%). The MIC range for *P. aeruginosa* and *E. coli* (≤0.03–1 g/mL) indicates a promising CFD activity profile against *Enterobacterales* and nonfermenting bacteria; higher values were recorded for *K. pneumoniae* and *A. baumannii* (≤0.03–4 g/mL and ≤0.03–> 6 4 g/mL, respectively). Both treatments lasted comparatively around 10 days, and the primary end-point defined as 14-day mortality rate was also similar among patients with VABP (15% vs. 13%). The 28-day mortality rate was similar in both groups among patients with VABP (23% vs. 22%). Clinical cure and microbiological eradication was achieved in 66% and 42% of cases for the CFD treated group and 56% and 34% for the MER group, respectively. The safeties of the CFD and MER therapies were also evaluated; the overall frequency of AEs was similar (88% vs. 86%), and the most common were UTIs, hypokalemia, diarrhea, and anemia. The preparation showed noninferior effectiveness in relation to MER; it also did not generate numerous severe TEAEs. In summary, it is a potential therapeutic option in the case of VABP with GNB etiology, including *A. baumannii* and CRE [107].

In a multicenter, randomized, and phase 3 study—CREDIBLE-CR—Bassetti et al. evaluated the efficacy of CFD therapy among patients with carbapenem-resistant GNB etiology infections. The study enrolled 152 patients who were randomly divided into the following two groups: those treated with CFD (N = 101, 2000 mg every 8 h) and those receiving BAT (N = 51). The percentages of patients presenting VABP were as follows: 24% vs. 27% in both groups. Importantly, the dominant pathogen isolated from patients suffering from VABP was CRAB (65% vs. 53%). The study showed a noninferior effect of CFD vs. BAT (primary end-point: clinical cure in TOC were 50% vs. 53% for the nosocomial infections group). Microbiological eradication in TOC accounted for 23% vs. 21%, respectively. Treatment-emergent adverse events occurred in 15% vs. 22%, respectively. Significantly, both 14- and 28-day mortalities were higher in the CFD vs. BAT groups (24% vs. 14% and 31% vs. 18%, respectively). This was probably due to the more frequent CRAB etiology of VABP/HAP among patients in the CFD-treated group than among those receiving BAT [108].

Falcone et al. evaluated the efficacy of CFD as a salvage therapy for the infection of nonfermenting NDM or CRE pathogens. The study included 10 patients, 4 of whom presented symptoms of VABP complicated by SARS-CoV2 infection. Each of the described cases had a different microbiological etiology, as follows: *A. baumannii*, *A. baumannii* + *K. pneumoniae* NDM, *K. pneumoniae* NDM, and *K. pneumoniae* NDM + *S. maltophila*. Significantly, three-quarters of patients showed clinical success with 30 days of therapy, and only one case of death was reported (30-day mortality ratio). In all cases, CFD monotherapy was performed at the site of previous combination regimens that did not cause clinical improvement in patients (including COL, ATM, TGC, CAZ-AVI, or FOS). The MIC values for *K. pneumoniae* NDM were in the range of 1–2 µg/mL in patients with VABP. The study demonstrated the benefits of CFD monotherapy as a salvage therapy for severe infections, e.g., VABP in patients who had exhausted other treatment options [109].

Another study by Falcone et al. compared the therapeutic efficacy of CFD vs. COL therapy in the treatment of infections with an etiology of CRAB. The study included 124 patients who were randomized into two groups: those treated with CFD (N = 47) and those treated with COL (N = 77). A total of 35 patients were clinically and laboratory confirmed to have a VABP infection with CRAB etiology (12 vs. 23 cases, respectively). Among the 12 patients treated with CFD (MIC = 0.12–2 mg/L), only 2 received the preparation as a monotherapy, and the others received CFD combined with TGC, FOS, or meropenem-vaborbactam (MER-VAB). The median duration of treatment with CFD vs. COL was comparable (between 15 and 13 days). The 30-day mortality ratio achieved a lower score among those treated with CFD vs. COL (34% vs. 54%, risk ratio of 0.44, 95% confidence interval of 0.22–0.66, *p* < 0.001) in the general population. Among patients treated for VABP, there was no significant difference in the 30-day mortality (58.3% vs. 56.5%). In terms of therapy safety, the advantage of CFD therapy in a COL-based regimen has been demonstrated. The incidence of TEAEs was higher in the COL-treated group at 21.1% (100% was AKI) vs. CFD at 2.1% (rash) [110].

Similar reports come from Rando et al. These researchers evaluated the efficacy of CFD therapy (mainly in combination therapy) in 13 patients with severe infections of multibacterial etiology including CRAB (10/13 developed VABP complicated by SARS-CoV2 infection). The median duration of treatment was 10 days, 7/13 patients achieved clinical cure, the remaining patients died despite treatment (30-day mortality ratio = 46%). No TEAEs associated with CFD treatment were reported. Although there was a considerable amount of mortality, it is noteworthy that the patients were critically ill, had numerous comorbidities, and the infection was multibacterial, complicated by SARS-CoV2 [111]. The effectiveness of various treatment options for VABP with the etiology of CRAB complicated by SARS-CoV2 infection in ICU was also evaluated. Russo et al. compared the efficacy of therapeutic regimens based on COL and CFD. The study enrolled 73 patients who were randomly assigned to appropriate subgroups treated with the appropriate drug regimen. All tested strains were classified as XDR or PDR. The population treated with CFD in combination with other drugs had lower 14- and 30-day mortality ratios than the population treated with COL-based regimens (5.2% and 31.5% vs. 75.9% and 98.1%, respectively). Special benefits in the 30-day survival ratio were attributed, especially, to the therapy with a combination of CFD + FOS [112].

Conclusions similar to those from Russo et al.’s study were derived from Rando et al.’s study, which compared the effectiveness of therapeutic regimens based on the combination of CFD + COL ± TGC or, among others, COL + TGC + FOS in VABP therapy in the etiology of CRAB complicated by SARS-CoV2 infection. The 28-day mortality ratio showed that the CFD-containing therapy was associated with a significantly lower risk of death than other regimens without CFD at 44% vs. 67%, respectively. As in the previous study, the significant benefits of CFD use in VABP with CRAB etiology in the field of increased patient survival were demonstrated [113]. The efficacy of CFD-based monotherapy and combination therapy in XDR and DTR (difficult-to-treat resistant) *P. aeruginosa* etiology infections was also evaluated. The study included 17 patients, and symptoms of VABP were presented by 7 patients (5/7 had SARS-CoV2 coinfection). Only one patient received CFD as monotherapy, the others were treated with combination therapy (COL inhaled, FOS, moxifloxacin, or CAZ-AVI). In 4/7 cases, the infection was caused by *P. aeruginosa*; in other cases, *K. pneumoniae* KPC, *S. maltophila*, and *A. baumannii* PDR was additionally found. In the VABP group, no significant TEAEs were found during treatment, and 3/7 patients died, while the remaining patients showed clinical success and microbiological eradication, with only 1/7 patients developing a recurrent infection. The study proves the effectiveness of CFD therapy in critically ill patients whose previous therapy, including TGC, CAZ-AVI, or MER, was unsuccessful [114].

In the single-center prospective observational study by Dalfino et al., different therapeutic regimens (including CFD or COL) were evaluated for CRAB etiology VABP suffered patients. The patient population (N = 90) was randomized into two groups: treated with a CFD-based regimen with inhaled COL (N = 40) and treated with a COL-based regimen with additional inhaled COL (N = 50). Clinical failure occurred less frequently in patients treated with CFD (25% vs. 48%). Microbiological failure was also shown more frequently among treated with COL combinations (30% vs. 60%). The 14-day mortality ratio was significantly lower for patients treated with CFD (10% vs. 38%). Of the CFD-based regimens, the highest percentage of clinical cures was recorded with the combination of CFD + FOS + COL inhaled [115].The findings are highly consistent with previously described studies [111,112,113].

The study by de la Fuente et al. evaluated the efficacy of CFD-based therapy in the treatment of carbapenem-resistant (CR) GNB infections in a population (N = 13) of patients hospitalized in an ICU. Eleven patients were confirmed clinically and in the laboratory to have VABP, showing the following etiological factors: 5/10 CR *P. aeruginosa*, 3/10 CR *Burkholderia cepacia*, 1/10 *S. maltophila*, and 1/10 CR *K. pneumoniae* KPC. Moreover, 4/10 patients received CFD alone, and the others were treated with CFD combination regimens with LEV/MER + TRM-STX/ciprofloxacin/COL/LEV + TRM-STX. In the group of patients treated for VABP, the 28-day mortality ratio reached 30%, and the remaining patients presented clinical success. All patients, except for one, developed microbiological eradication. According to the study, the treated group did not present severe TEAEs, and the high effectiveness of VABP therapy with CFD for a CR-GNB etiology was demonstrated [116].

A summary of the clinical efficiency of the cefiderocol therapy among patients with VABP is presented in Table 9.

#### 5.1.3. Case Studies

Mercadante et al. described the use of CFD as a salvage therapy in an 8-month-old infant treated for VABP with an etiology of *P. aeruginosa* VIM. The patient’s severe condition led to the need for therapy with ECMO (extra-corporeal membrane oxygenation) and the use of CRRT (continuous-renal-replacement Therapy). The previous therapy, which included CAZ-AVI + ATM, did not lead to an improvement. After a week of treatment, the patient’s condition deteriorated. Because of the sensitivity of *P. aeruginosa* to CFD (MIC = 1 µg/mL), the decision was made to use the off-label preparation. The therapy lasted for 14 days and led to the patient’s recovery; after 4 days from the beginning of treatment, the condition improved so much that ECMO and CRRT were discontinued, and after a week of treatment, negative blood and BAL (bronchoalveolar lavage) cultures were obtained. The study demonstrates the effectiveness of CFD-based salvage therapy in infants undergoing invasive intensive care procedures, as follows: ECMO and CRRT [117].

Another case study presented the successful use of CFD in the treatment of necrotizing pneumonia caused by *P. aeruginosa* associated with mechanical ventilation. The pathogen developed resistances to CEF-TAZ and MER during previous antibiotic therapy. The combination of CFD with TOB and ciprofloxacin led to the patient’s recovery and confirmed microbiological eradication. The evidence suggests that CFD is effective in salvage therapy for people with VABP caused by CEF-TAZ-resistant MDR P. aeruginosa [118].

#### 5.1.4. Mechanism of Action of β-Lactam Drugs

The mechanism of action of the cephalosporins described in this manuscript is to block the final stage of cell wall biosynthesis. These antibiotics inhibit the activity of transpeptidases (PBPs)—protoplasmic proteins responsible for cross-linking the bacterial cell wall. As a result of their action, bonds among its components—mucopolysaccharides—are not formed, causing lysis of the bacterial cell.

The combinations we looked at closely in this review increase the sensitivity of strains to the associated cephalosporin. This is due to the fact that β-lactamase inhibitors bind to some PBPs, thus protecting the antibiotic from enzymatic degradation in the presence of β-lactamase-producing bacteria. In addition, the higher the class of cephalosporins used in combination with the inhibitor, the broader the activity against β-lactamases. Therefore, wider use in the treatment of infections with a G-negative bacterial etiology in order from the least to most active is as follows: CFP-SBT, CEF-TAZ, and CAZ-AVI.

The highest activity is shown by cefiderocol, a siderophore cephalosporin, which has been confirmed against WHO priority pathogens, as follows: *A. baumannii*, *P. aeruginosa*, and *Enterobacteriaceae* resistant to carbapenems. Moreover, its efficacy against ESβL-, AmpC-, KPC-, and MBL-positive strains was confirmed. The mechanism of cefiderocol is based on the inhibition of the synthesis of the bacterial cell wall by binding to PBP proteins, mainly PBP3. This cephalosporin, in the first step, binds iron, which is necessary for bacteria to function, and then, by active transport, enters the periplasmic space. During this process, it reduces the number of porins in the outer membrane and the expression of MDR pumps responsible for the efflux [100]. 

## 6. Material and Methods

This review refers to articles from the Scopus, Web of Science, PubMed and Google Scholar databases. In total, 115 papers were included. The following keywords were used in the search of the titles and abstracts of the articles: ‘ventilator-associated pneumonia’, ‘ceftolozane tazobactam’, ‘ceftazidime avibactam’, ‘cefoperazone sulbactam’, and ‘cefiderocol’. Table 1 presents ability of cefiderocol and selected cephalosporin-inhibitor combinations to overcome multidrug resistance among bacteria. Table 3, Table 5, Table 7, and Table 9 refer to the clinical efficiency of the described preparations in VABP therapy. Table 2, Table 4, Table 6, and Table 8 show the in vitro antimicrobial activity of the mentioned drugs against clinical isolates of bacteria derived from patients suffering from VABP. Figure 1, Figure 2, Figure 3 and Figure 4 present two-dimensional structures of the revealed preparations. Figure 5 depicts the scheme of the molecular action of the described β-lactams inside the bacterial cell.

## 7. Conclusions

New combinations of cephalosporins with β-lactamase inhibitors are a useful therapeutic option in the treatment of serious nosocomial infections, e.g., HAP and VABP. The addition of β-lactamase inhibitors extend the spectrum of action of classical β-lactam antibiotics by bacteria with a high potential of resistance, especially KPC and ESβL *Enterobacteriaceae*. The analyzed clinical trials report that the BL + BLI combination scan be used to optimize treatment times, minimize the risk of adverse events, and improve patient survival. The use of the described pharmaceuticals allowed, in many cases, for the limitation of the use of multidrug combination therapies based on colistin, a last resort antibiotic, which is connected with a high risk of nephrotoxicity. In the studies mentioned above, it was shown that the drugs used obtained a generally noninferior profile of action compared to polymyxins or carbapenems, which is an important premise in the era of increasing antibiotic resistance, especially in ICU wards. Special hopes are, therefore, raised by the new generation of β-lactam antibiotics: siderophore cephalosporins and their representative—cefiderocol. In both in vitro studies and clinical trials, the preparation displayed an impressive level of action, demonstrating that it has therapeutic efficacy comparable or superior to colistin and meropenem. Particularly important is the high activity of cefiderocol against resistance to the carbapenems *A. baumannii* and *S. maltophila* (pathogens especially dangerous to critically ill ICU patients), which has been repeatedly demonstrated in clinical trials. In conclusion, while VABP is still a challenge for clinicians, the cefiderocol and BL + BLI combinations described above can be used as salvage therapy in patients treated in the ICU ward when the other treatment options have been exhausted.

Limited treatment options for patients developing VABP should also encourage the search for innovative treatments for this life-threatening infection. Because of the increase in antibiotic resistance, it is necessary to develop products that can combine the properties of classic drugs (e.g., inhibition of PBP protein) with a novel mechanism to break down antibiotic resistance (e.g., siderophore mechanism). Future directions of research should oscillate around the search for new ways of influencing bacterial cells. High hopes in this field may be raised by substances of natural origin (e.g., plant extracts and alkaloids), which due to their cytostatic properties have already been used in, among others, the therapy of oncological diseases. It also seems particularly important to search for new combinations of classic antibiotics with antibacterial substances of natural origin. The detection of additive or synergistic interactions among these compounds could reduce the use of last resort drugs (e.g., colistin) in favor of preparations with, among others, lower toxicity. Although more research is needed, the combination of innovative therapeutic strategies and rapid molecular diagnostics systems (MALDI-TOF MS) is a worthwhile option in the face of the growing global problem of antibiotic resistance.

However, this study has some limitations. Various clinical studies had significant differences, including the number of VABP patients in each cohort, which could lead to different statistical power values and, sometimes, bias in the outcomes. In addition, individuals who qualified for specific groups and received a particular treatment regimen could differ disproportionately based on their severity of concomitant diseases, age, or the number of previous hospitalizations (a factor that may increase the risk of colonization with MDR pathogens). The treatment effects may have been different, although not due to drug action directly but to the worsening prognosis of specific patients. It was also difficult to determine if the clinical success was caused by a specific drug or combined preparations in multiple situations in which the evaluated drugs were used in a polytherapy. The antibiotic resistance profile of pathogens, which are the underlying causes of VABP, can be different depending on the patient’s location or the type of ward they are in during hospitalization. Conducting a meta-analysis study, which would enable us to develop the topic in depth, seems worthwhile for the described reasons.

## Figures and Tables

**Figure 1 antibiotics-13-00445-f001:**
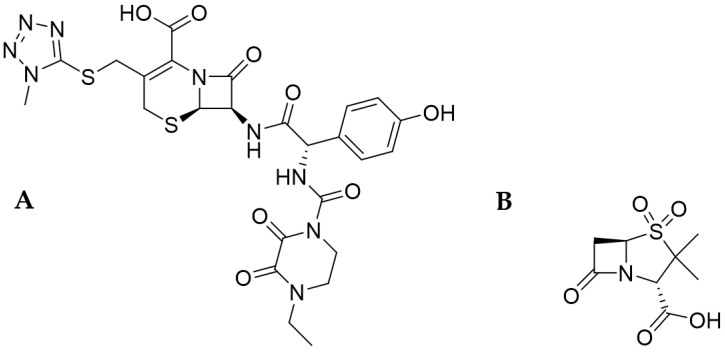
Structural models of cefoperazone (**A**) and sulbactam (**B**).

**Figure 2 antibiotics-13-00445-f002:**
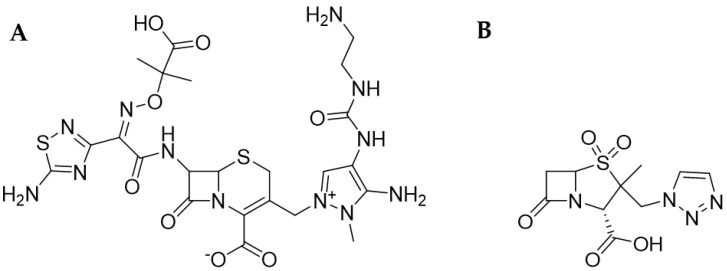
The structural models of ceftolozane (**A**) and tazobactam (**B**).

**Figure 3 antibiotics-13-00445-f003:**
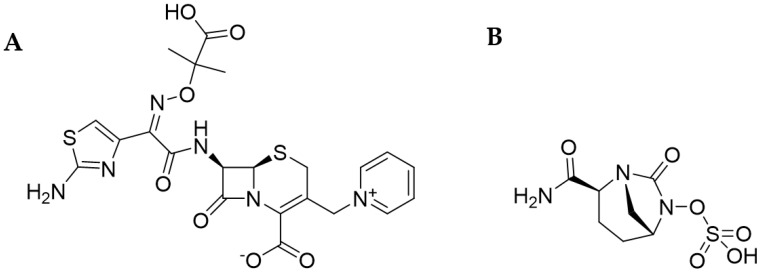
The structural models of ceftazidime (**A**) and avibactam (**B**).

**Figure 4 antibiotics-13-00445-f004:**
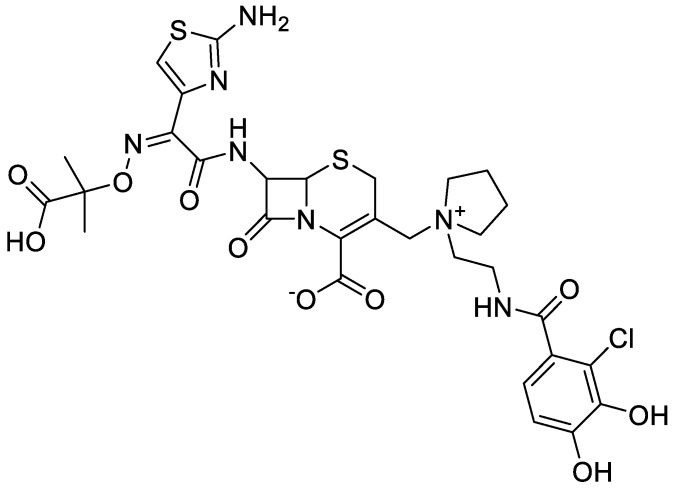
The structural model of cefiderocol.

**Figure 5 antibiotics-13-00445-f005:**
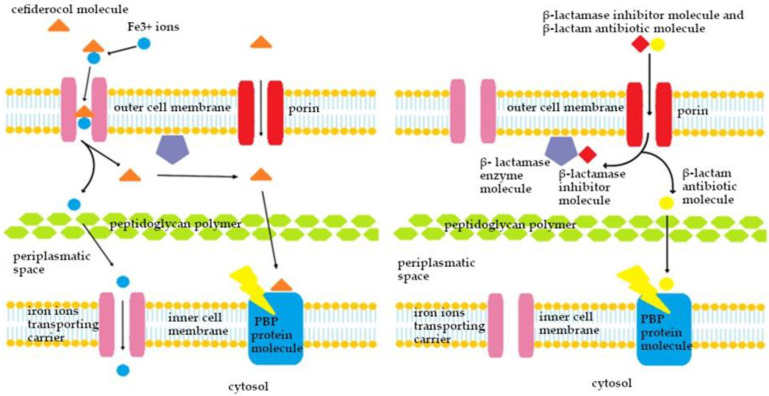
Comparison of the molecular mechanism of action of cefiderocol and the β-lactam β-lactamase inhibitor combination on the bacterial cell wall. Inhibition of peptidoglycan synthesis due to β-lactam antibiotic molecule’s interaction with PBP is marked with a lightning bolt symbol.

**Table 1 antibiotics-13-00445-t001:** Ability of siderophore cephalosporins and selected cephalosporin-inhibitor combinations to overcome multidrug resistance among bacteria.

Cephalosporin-Inhibitor Combination/Antibiotic	Spectrum of Action According to Ambler Classification	Examples of β-Lactamases	References
Cefoperazone-sulbactam	class A	narrow spectrum: TEM-1 and TEM-2	[18]
	class C	AmpC	
Ceftolozane-tazobactam	class A	extended spectrum (ESβL): SHV-2 and CTX-M-15	[19,20]
	class C	AmpC	
Ceftazidime-avibactam	class A	narrow spectrum: TEM-1	[21,22]
		extended spectrum: SHV and CTX-M	
		KPC-2 and KPC-3	
	class C	AmpC	
	class D	OXA-48	
Cefiderocol	class A	extended spectrum, e.g., SHV type	[23]
		KPC	
	class B	MBL: VIM, IMP, and NDM	
	class C	AmpC	
	class D	OXA-48 and OXA-23	

AmpC—cephalosporinases encoded on the chromosomes of many of the *Enterobacteriaceae*; CTX-M-15—CTX-M-type ESβL; ESβL—extended-spectrum β-lactamase; KPC—*K. pneumoniae* carbapenemase; IMP—metallo-β-lactamase type imipenemase; MBL—metallo-β-lactamase; NDM—New Delhi MBL; OXA—oxacillinase-type β-lactamase; SHV—sulfhydryl variable penicillinase; TEM—plasmid-encoded β-lactamase in Gram-negative bacteria; VIM—Verona integron-encoded MBL.

**Table 2 antibiotics-13-00445-t002:** In vitro antimicrobial activity of CFP-STB and its combinations against VABP clinical isolates.

Pathogen	Antibiotic Scheme Used In Vitro	MIC Value	Reference
MSSA	CFP-SBT	MIC_50_ = 2 µg/mL	[39]
	CFP	MIC_50_ = 4 µg/mL	
*K. pneumoniae*	CFP-SBT	MIC_90_ = 32 µg/mL	
	CFP	MIC_90_ = 128 µg/mL	
*P. aeruginosa*	CFP-SBT	MIC_90_ = 32 µg/mL	
	CFP	MIC_90_ = 128 µg/mL	
*A. baumannii*	CFP-SBT	MIC_90_ = 64 µg/mL	
	CFP	MIC_90_ > 128 µg/mL	
*S. maltophila*	CFP-SBT	MIC_50_ = 64 µg/mL	
	CFP	MIC_50_ = 128 µg/mL	
MDR *P. aeruginosa*	CFP-SBT	MIC_50_ = 64 µg/mL	[42]
		MIC_90_ = 128 µg/mL	
	CFP-SBT + AZT	MIC_50_ = 16 µg/mL	
		MIC_90_ = 64 µg/mL	
Carbapenem-resistant *A. baumannii*	CFP-SBT + MER	MIC_50_ = 16 µg/mL	[37]
		MIC_90_ = 64 µg/mL	

AZT—azithromycin; CFP-SBT—cefoperazone-sulbactam; MDR—multidrug-resistant; MER—meropenem; MSSA—methicillin-susceptible *S. aureus*.

**Table 3 antibiotics-13-00445-t003:** Efficacy of CFP-SBT in the treatment of VABP.

Number of Patients	Treatment Scheme in VABP Population	Outcomes of Trial			Etiology of Infection	Reference
N = 166	CFP-SBT (N = 79)	CC:		73.1%	*P. aeruginosa* *A. baumannii*	[43]
	CPM (N = 87)			56.8%		
N = 317	CFP-SBT (N = 167)	CC		85%	*K. pneumoniae* *P. aeruginosa* *A. baumannii*	[42]
				81%		
	PIP-TAZ (N = 150)	IHMR		24%		
				23%		
N = 65	CFP-SBT (N = 37)	CC in VABP group:	78.4%		*A. baumannii*	[44]
			71.4%		*P. aeruginosa*	
	PIP-TAZ (N = 28)	Total mortality ratio:	13.5%		*K. pneumoniae*	
			3.6%		*P. aeruginosa*	
N = 308	CFP-SBT (N = 154)	CC:	50%		MDR (70%): *K. pneumoniae* *E. coli* *P. aeruginosa* *A. baumannii*	[30]
			51.2%			
		14-MR:	29.2%			
	PIP-TAZ (N = 154)		35%			
		28-MR:	46.1%			
			42.8%			
N = 42	CFP-SBT + TGC	CC:	85.7%		XDR *A. baumannii*	[46]
	TGC		47.6%			
N = 80	Combination therapy with CFP-SBT (N = 52)	14-MR	17%		Carbapenem-resistant *A. baumannii*	[47]
			39%			
		30-MR	35%			
			61%			
	Combination therapy without CFP-SBT (N = 38)	IHMR	39%			
			68%			

14/28/30-MR—14/28/30-day mortality ratio; CC—clinical cure; CFP-SBT—cefoperazone-sulbactam; CPM—cefepime; IHMR—in-hospital mortality ratio; MDR—multidrug-resistant; PIP-TAZ—piperacillin-tazobactam; TGC—tigecycline; VABP—ventilator-associated bacterial pneumonia; XDR—extensively drug-resistant.

**Table 4 antibiotics-13-00445-t004:** In vitro antimicrobial activity of CEF-TAZ and its combinations against VABP clinical isolates.

Pathogen	Antibiotic Scheme Used In Vitro	MIC Value	Reference
XDR *P. aeruginosa*	CEF-TAZ	MIC_50_ = 2 mg/mL MIC_90_ > 32 mg/mL	[58]
		MIC_50_ = 1 mg/mL MIC_90_ = 8 mg/mL	
MDR *P. aeruginosa*	CEF-TAZ	MIC_50_ = 1 mg/L	[19]
		MIC_90_ = 8 mg/L	
XDR *P. aeruginosa*	CEF-TAZ	MIC_50_ = 2 mg/L	
		MIC_90_ = 16 mg/L	
Non-CRE *E. coli*	CEF-TAZ	MIC_50_ = 0.5 mg/L	
		MIC_90_ = 2 mg/L	
Non-CRE *K. pneumoniae*	CEF-TAZ	MIC_50_ = 1 mg/L	
		MIC_90_ = 8 mg/L	
ESβL non-CRE *Enterobacterales*	CEF-TAZ	0.5–8 mg/L	[59]
*P. aeruginosa*	CEF-TAZ	MIC_50_ = 0.5 mg/L	[65]
		MIC_90_ = 8 mg/L	
*K. pneumoniae*	CEF-TAZ	MIC_50_ = 0.5 mg/L	
		MIC_90_ > 64 mg/L	

CEF-TAZ—ceftolozane-tazobactam; CRE—carbapenem-resistant *Enterobacterales;* MDR—multidrug-resistant; XDR—extensively drug-resistant.

**Table 5 antibiotics-13-00445-t005:** Efficacy of CEF-TAZ in the treatment of VABP.

Number of Patients	Treatment Scheme in VABP Population	Outcomes of Trial		Etiology of Infections	Reference
N = 726 (71% VABP)	CEF-TAZ (N = 362)	28-MR	24%	*K. pneumoniae*, *E. coli* *P. aeruginosa* (including ESβL)	[66]
			25.3%		
		Clinical response at TOC	54%		
	MER (N = 364)		53%		
		Microbiological eradication ratio	73.1%		
			68%		
N = 200 (52% VABP)	CEF-TAZ (N = 100)	CC	81%	MDR/XDR *P. aeruginosa*	[69]
			61%		
	Aminoglycosides (TOB, GEN, AKC)/COL (N = 100)	IHMR	20%		
			25%		
N = 51	CEF-TAZ (N = 18)	CC	72.2%	XDR *P. aeruginosa*	[70]
			30.3%		
		Microbiological eradication ratio	44.4%		
			15.2%		
	COL (N = 33)	28-MR	27.8%		
			33.3%		
		Frequency of AEs	55.5%		
			72.7%		
N = 205 (63/205 VABP)	CEF-TAZ (N = 63)	30-MR	35%	MDR *P. aeruginosa*	[73]
		CC	50%		
		Microbiological eradication in the EOT	53.4%		
N = 206 (46.6% VABP)	CEF-TAZ (N = 118)	Clinical failure	23.7%	MDR/XDR *P. aeruginosa*	[74]
			48.9%		
		30-MR	15.3%		
	BAT (N = 88)		20.5%		
		Frequency of TEAEs	10.2%		
			33%		

28/30-MR—28/30-day mortality ratio; AE—adverse event; AKC—amikacin; BAT—best available treatment (therapy included combinations of, among others, piperacillin-tazobactam, cefepime, meropenem, ceftazidime-avibactam, ceftazidime, and colistin); CC—clinical cure; CEF-TAZ—ceftolozane-tazobactam; COL—colistin; EOT—end of treatment; GEN—gentamicin; IHMR—in-hospital mortality ratio; MDR—multidrug-resistant; MER—meropenem; TEAE—treatment emergent adverse event; TOB—tobramycin; TOC—test of cure; VABP—ventilator-associated bacterial pneumonia; XDR—extensively drug-resistant.

**Table 6 antibiotics-13-00445-t006:** In vitro antimicrobial activity of CAZ-AVI and its combinations against VABP clinical isolates.

Pathogens	Antibiotic Scheme Used In Vitro	MIC Value	Reference
KPC *Enterobacteriaceae*	CAZ-AVI	MIC_50_ = 0.5 mg/L	[82]
		MIC_90_ = 2 mg/L	
CRE *Enterobacteriaceae*		MIC_50_ = 0.5 mg/L	
		MIC_90_ = 2 mg/L	
*P. aeruginosa*		MIC_50_ = 2 mg/L	
		MIC_90_ = 4 mg/L	
*P. aeruginosa* (MER-NS, CAZ-NS, or MDR strains)		MIC_50_ = 4 mg/L	
		MIC_90_ = 16 mg/L	
XDR *P. aeruginosa*		MIC_50_ = 8 mg/L	
		MIC_90_ = 32 mg/L	
*A. baumannii*		MIC_50_ = 16 mg/L	
		MIC_90_ > 32 mg/L	
MDR *P. aeruginosa*	CAZ-AVI	MIC_50_ = 2 mg/L	[57]
		MIC_90_ = 16 mg/L	
KPC-2 *K. pneumoniae*	CAZ-AVI	MIC = 0.4–0.8 mg/L	[83]
OXA-232 *K. pneumoniae*		MIC = 0.2 mg/L	
NDM *K. pneumoniae*		MIC = 0.5–256 mg/L	
KPC-2 + NDM *K. pneumoniae*		MIC = 8–128 mg/L	
MDR *P. aeruginosa*	CAZ-AVI	MIC_50_ = 8 mg/L	[84]
		MIC_90_ = 64 mg/L	
MDR *K. pneumoniae*		MIC_50_ = 0.5 mg/L	
		MIC_90_ = 1 mg/L	
*P. aeruginosa*	CAZ-AVI	MIC_50_ = 2 mg/L	[86]
		MIC_90_ = 4 mg/L	
MER-NS *P. aeruginosa*		MIC_50_ = 4 mg/L MIC_90_ = 16 mg/L	
PIP-TAZ-NS *P. aeruginosa*			
MDR *P. aeruginosa*			
XDR *P. aeruginosa*		MIC_50_ = 8 mg/L	
		MIC_90_ = 32 mg/L	
*A. baumannii*		MIC_50_ = 16 mg/L	
		MIC_90_ > 32 mg/L	
ESβL *K. pneumoniae*		MIC_50_ = 0.25 mg/L	
		MIC_90_ = 1 mg/L	
ESβL *E. coli*		MIC_50_ = 0.12 mg/L	
		MIC_90_ = 0.5 mg/L	

CAZ-AVI—ceftazidime-avibactam; CRE—carbapenem-resistant *Enterobacterales*; MDR—multidrug-resistant; MER-NS—meropenem-nonsusceptible; PIP-TAZ-NS—piperacillin-tazobactam-nonsusceptible; XDR—extensively drug-resistant.

**Table 7 antibiotics-13-00445-t007:** Efficacy of CAZ-AVI in the treatment of VABP.

Number of Patients	Treatment Scheme in VABP Population	Outcomes of Trial		Etiology of Infection	Reference
N = 726	CAZ-AVI (N = 356) (33% VABP)	Clinical success in mITT population	70.3%	*K. pneumoniae* *P. aeruginosa*	[87]
			74.2%		
		Clinical success in CEP population	77.5%		
			75.9%		
	MER (N = 370) (35% VABP)	28-MR	9%		
			7%		
		Frequency of AEs	75%		
			74%		
N = 105 (71.4% VABP)	CAZ-AVI (N = 43)	Clinical success	51.2%	CRE *K. pneumoniae*	[89]
			29%		
		Microbiological success	74.4%		
	TGC (N = 62)		33.9%		
		28-MR	69.8%		
			66.1%		
N = 77 (33.8% VABP)	CAZ-AVI (N = 41, 19/41 VABP)	CC	80.5%	CRE: *P. aeruginosa* *K. pneumoniae* *E. coli*	[90]
			52.8%		
		Microbiological eradication ratio	94.3%		
	BAT (N = 36, 7/36 VABP)		67.7%		
		28-day survival ratio	85.4%		
			61.1%		
N = 62	CAZ-AVI monotherapy (N = 21)	30-MR	47.6%	CRE: *P. aeruginosa* *K. pneumoniae* *E. coli*	[92]
			24.4%		
	CAZ-AVI combined therapy (N = 41)	Microbiological eradication ratio	42.9%		
			61%		
N = 61	CAZ-AVI in monotherapy 53% CAZ-AVI in combined therapy 47% (scheme obtained, e.g., COL/MER/ATM)	CC by the day 14	54.1%	*P. aeruginosa*: MDR—91.8% XDR—8.2%	[95]
		30-MR	13.1%		
		Recurrence by the day 90	12.5%		
		30-day survival ratio	93.8% (for CAZ-AVI monotherapy)		
N = 203 (37.4% LRTI include VABP)	CAZ-AVI monotherapy (N = 203)	Clinical success	70.9%	CRE: *P. aeruginosa* *K. pneumoniae*	[96]
		Recurrence by the day 30 ratio	5.9%		
		30-MR	17.2%		

28/30-MR—28/30-day mortality ratio; AE—adverse event; ATM—aztreonam; BAT—best available therapy; CAZ-AVI—ceftazidime-avibactam; CC—clinical cure; CEP—clinical evaluable population (mITT patients after receiving treatment); COL—colistin; CRE—carbapenem-resistant *Enterobacterales;* GNB—Gram-negative bacteria; LRTI—lower respiratory tract infection; MDR—multidrug-resistant; MER—meropenem; mITT—modified intention to treat population (patients meeting entry trial criteria with confirmed Gram (−) bacteria etiology infection); TGC—tigecycline; VABP—ventilator-associated bacterial pneumonia; XDR—extensively drug-resistant.

**Table 8 antibiotics-13-00445-t008:** In vitro antimicrobial activity of CFD and its combinations against VABP clinical isolates.

Pathogens	Antibiotic Scheme Used In Vitro	MIC Value	References
*K. pneumoniae*	CFD	MIC_90_ = 1 µg/mL	[20,104]
*P. aeruginosa*		MIC_90_ = 0.5 µg/mL	
*A. baumannii*		MIC_90_ = 1–4 µg/mL	
*S. maltophila*		MIC_90_ = 0.25–0.5 µg/mL	
MER-NS *P. aeruginosa*		MIC_90_ = 0.5–1 µg/mL	
MER-NS *A. baumannii*		MIC_90_ = 1–4 µg/mL	
carbapenem-NS *Enterobacteriaceae*	CFD	MIC_90_ = 4 µg/mL	[105]
*P. aeruginosa*	CFD	MIC_50_ = 0.12 mg/L	[106]
		MIC_90_ = 0.5 mg/L	
XDR *P. aeruginosa*		MIC_50_ = 0.12 mg/L	
		MIC_90_ = 1 mg/L	
MER-NS *P. aeruginosa*		MIC_50_ = 0.12 mg/L	
		MIC_90_ = 1 mg/L	
*A. baumannii*		MIC_50_ = 0.25 mg/L	
		MIC_90_ = 1 mg/L	
MER-NS *A. baumannii*		MIC_50_ = 0.5 mg/L	
		MIC_90_ = 2 mg/L	
*S. maltophila*		MIC_50_ = 0.12 mg/L	
		MIC_90_ = 0.5 mg/L	

CFD—cefiderocol; MER-NS—meropenem-nonsusceptible; XDR—extensively drug-resistant.

**Table 9 antibiotics-13-00445-t009:** Efficacy of CFD in the treatment of VABP.

Number of Patients	Treatment Scheme in VABP Population	Outcomes of Trial		Etiology of Infection	Reference
N = 300	CFD (N = 148) (41% VABP)	14-MR	15%	*K. pneumoniae* *P. aeruginosa* *A. baumannii* (CRE strains: 70% ESβL producers: 43–67%)	[107]
			13%		
		28-MR	23%		
			22%		
	MER (N = 152) (44% VABP)	CC	66%		
			56%		
		Microbiological eradication	42%		
			34%		
N = 152	CFD (N = 101) (24% VABP)	CC at TOC	50%	CR-GNB With a predominance of CRAB (65% of isolates in the CFD and 53% in the BAT groups)	[108]
			53%		
		Microbiological eradication at TOC	23%		
			21%		
	BAT (N = 51) (27% VABP)	14-MR	24%		
			31%		
		28-MR	14%		
			18%		
N = 35 (VABP group)	12/35 CFD-based therapy	30-MR	58.3%	CRAB	[110]
	23/35 COL-based therapy		56.5%		
N = 73	CFD (N = 19)-based therapy	14-MR	5.2%	CRAB complicated with SARS-CoV2	[112]
			75.9%		
	COL (N = 54)-based therapy	28-MR	31.5%		
			98.1%		
N = 90	CFD-based therapy with additional inh. COL (N = 40)	Clinical failure	25%	CRAB	[115]
			48%		
		Microbiological failure	30%		
	COL-based therapy with additional inh. COL (N = 50)		60%		
		14-MR	10%		
			38%		

14/28/30-MR—14/28/30-day mortality ratio; CC—clinical cure; CFD—cefiderocol; COL—colistin; CRAB—carbapenem-resistant *A. baumannii*; CRE—carbapenem-resistant *Enterobacterales*; CR-GNB—carbapenem-resistant Gram negative bacteria; inh.—inhaled; MER—meropenem; TOC—test of cure; VABP—ventilator-associated bacterial pneumonia.

## Data Availability

Not applicable.

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
