# Peer review of "Novel Siderophore Cephalosporin and Combinations of Cephalosporins with β-Lactamase Inhibitors as an Advancement in Treatment of Ventilator-Associated Pneumonia"

_antibiotics, 2024, doi:10.3390/antibiotics13050445_

Round 1

Reviewer 1 Report

Comments and Suggestions for Authors

This review addresses the growing challenge of treating ventilator-associated pneumonia caused by multidrug-resistant and extensively drug-resistant Gram-negative bacteria. It highlights the critical need for novel pharmacotherapies, particularly highlighting the promise of new β-lactam antibiotics such as siderophore cephalosporins combined with β-lactamase inhibitors. These innovative treatments are proposed due to their unique mechanisms of action, aim to circumvent current drug resistances. The review concludes by summarizing the results from clinical trials, in vitro studies, and case reports that suggest these new therapeutic approaches may provide equal or enhanced clinical efficacy compared to traditional treatments, with fewer adverse events. Here are few suggestions,

Lane 13: Remove the extra space in "patients hospitalized".

Lane 14: Replace "B-lactams" with "β-lactams".

Tables 1 and 2: Add a period at the end of each title.

Lane 364: Correct "Figure 2B" to "Figure 2A".

Lane 382: Correct the subtitle from "2.2" to "2.1".

Lane 453: Please explain the numbers in the brackets, fluoroquinolone (1) and aminoglycoside (2).

Lane 464: Add a space between "counted" and "63".

Table 4: Add a period at the end of the title.

Lane 562: Remove the extra space in "study also".

Figure 3: Ensure there is a description in the manuscript.

Lane 650: Remove the capitalization for "Ceftazidime-Avibactam".

Lane 688: Include references for the study mentioned.

Figure 4: Ensure there is a description in the manuscript.

Table 6: Add a period at the end of the title.

Lane 935: Insert a comma between "A. baumannii + K. pneumoniae NDM" and "K. pneumoniae NDM".

Lane 1008: Remove the extra space in "KPC .".

Table 8: Add a period at the end of the title.

Conclusions: It's important to discuss how each acknowledged limitation might impact the interpretation of the findings. This could involve specifying potential biases, variances in data collection, or external influences that could affect the study's outcomes. Suggestions for how future research could address these limitations can strengthen the conclusions, offering paths for deeper investigation or alternative methodologies.

Overall, the manuscript could be improved by providing more detail on the mechanism of action for the discussed treatments, particularly how the biochemical interactions within bacterial cells contribute to the drugs effectiveness against specific pathogens. This detail will enrich readers understanding and underscore the innovations in pharmacotherapy for resistant bacterial infections.

Comments on the Quality of English Language

1. Consistency in Tense:

The manuscript switches between past and present tense inconsistently. 

2. Comma Usage:

Several sentences are missing commas, which could lead to misinterpretation of the information.

Author Response

Dear Reviewer,

Thank you very much for studying the topics undertaken in our Article and for your detailed review. In correcting our Review, we followed all your recommendations, for which we are very thankful. Our corrections are presented below:

Lane 13: Remove the extra space in "patients hospitalized".

The space has been removed.

Lane 14: Replace "B-lactams" with "β-lactams".

Replaced with consideration of correct spelling.

Tables 1 and 2: Add a period at the end of each title.

A period was added in the title of the Tables.

Lane 364: Correct "Figure 2B" to "Figure 2A".

Replaced with correct numbers.

Lane 382: Correct the subtitle from "2.2" to "2.1".

Subtitle numbers have been replaced.

Lane 453: Please explain the numbers in the brackets, fluoroquinolone (1) and aminoglycoside (2).

Numbers were removed as they were confusing.

Lane 464: Add a space between "counted" and "63".

Space has been added.

Table 4: Add a period at the end of the title.

A period was added in the title of the Table.

Lane 562: Remove the extra space in "study also".

The space has been removed.

Figure 3: Ensure there is a description in the manuscript.

We actually omitted the description of Figure 3 from the text. Thank you for pointing this out.

Lane 650: Remove the capitalization for "Ceftazidime-Avibactam".

Capital letters have been changed to lowercase.

Lane 688: Include references for the study mentioned.

A citation from Tsolaki et al. (2020) has been added to the studies discussed.

Figure 4: Ensure there is a description in the manuscript.

We actually omitted the description of Figure 3 from the text. Thank you for pointing this out.

Table 6: Add a period at the end of the title.

A period was added in the title of the Table.

Lane 935: Insert a comma between "A. baumannii + K. pneumoniae NDM" and "K. pneumoniae NDM".

The comma has been inserted in the correct place. 

Lane 1008: Remove the extra space in "KPC .".

The extra space has been removed.

Table 8: Add a period at the end of the title.

A period was added in the title of the Table.

Conclusions: It's important to discuss how each acknowledged limitation might impact the interpretation of the findings. This could involve specifying potential biases, variances in data collection, or external influences that could affect the study's outcomes.

Limitations affecting the results of the studies that we recognized in describing the research presented in this Review are listed in "Conclusions." This part reads as follows:

„However, this study has some limitations. Various clinical studies had significant differences, including the number of VABP patients in each cohort, which could lead to different statistical power values and sometimes bias in the outcomes. In addition, individuals who were qualified for specific groups and received a particular treatment regimen could differ disproportionately based on their severity of concomitant diseases, age, or the number of previous hospitalizations (a factor that may increase the risk of colonization with MDR pathogens). The treatment effects may have been different, although not due to drug action directly, but to the worsening prognosis of specific patients. It was also difficult to determine if the clinical success was caused by a specific drug or combined preparations in multiple situations when evaluated drugs were used in a polytherapy. The antibiotic resistance profile of pathogens, which were the underlying causes of VABP, can be different depending on the patient's location or the type of ward they were in during hospitalization. Conducting a meta-analysis study which would enable us to develop the topic in depth seems worthwhile due to described reasons.”

Suggestions for how future research could address these limitations can strengthen the conclusions, offering paths for deeper investigation or alternative methodologies.

Possible future research directions are included in the conclusion of this Review. This fragment reads:

”Limited treatment options for patients developing VABP should also encourage the search for innovative treatments for this life-threatening infection. Due to the increase in antibiotic resistance, it is necessary to develop products that can combine the properties of classic drugs (e.g. inhibition of PBP protein) with a novel mechanism to break down antibiotic resistance (e.g. siderophore mechanism). Future directions of research should oscillate around the search for new ways of influencing bacterial cells. High hopes in this field may be raised by substances of natural origin (e.g. plant extracts and alkaloids), which due to their cytostatic properties have already been used in, among others, the therapy of oncological diseases. It also seems particularly important to search for new combinations of classic antibiotics with antibacterial substances of natural origin. Detection of additive or synergistic interactions between these compounds could reduce the use of last resort drugs (colistin) in favour of preparations with, among others, lower toxicity.  Although more research is needed, the combination of innovative therapeutic strategies and rapid molecular diagnostics systems (MALDI-TOF MS) is a worthwhile option in the face of the growing global problem of antibiotic resistance.”

Overall, the manuscript could be improved by providing more detail on the mechanism of action for the discussed treatments, particularly how the biochemical interactions within bacterial cells contribute to the drugs effectiveness against specific pathogens. This detail will enrich readers understanding and underscore the innovations in pharmacotherapy for resistant bacterial infections.

As recommended by the Reviewer, the Review has been improved with an additional section that briefly discusses the mechanisms of action of the antibiotics described in the Manuscript and their combinations with β-lactamase inhibitors. Its contents are presented below:

”The mechanism of action of the cephalosporins described in following Manuscript is to block the final stage of cell wall biosynthesis. These antibiotics inhibit the activity of transpeptidases (PBPs) - protoplasmic proteins responsible for cross-linking the bacterial cell wall. As a result of their action, bonds between its components - mucopolysaccharides - are not formed, causing lysis of the bacterial cell.

The combinations we looked at closely in this Review increase the sensitivity of strains to the associated cephalosporin. This is due to the fact that β-lactamase inhibitors bind to some PBPs, thus protecting the antibiotic from enzymatic degradation in the presence of β-lactamase-producing bacteria. In addition, the higher the class of cephalosporins used in combination with the inhibitor, the broader the activity against β-lactamases. And therefore wider use in the treatment of infections with G-negative bacterial etiology in order from least to most active: CFP-SBT, CEF-TAZ and CAZ-AVI.

The highest activity is shown by cefiderocol, a siderophore cephalosporin, which has been confirmed against WHO priority pathogens, i.e. A. baumannii, P. aeruginosa and Enterobacteriaceae resistant to carbapenems. Moreover, its efficacy against ESBL-, AmpC-, KPC-, and MBL-positive strains was confirmed. The mechanism of cefiderocol is based on the inhibition of the synthesis of the bacterial cell wall by binding to PBP proteins, mainly PBP-3. This cephalosporin in the first step binds iron, necessary for bacteria to function, and then by active transport enters the periplasmic space. During this process, it reduces the number of porins in the outer membrane and the expression of MDR pumps responsible for the efflux (Sato 2019). 

In addition, the Manuscript was reviewed by a person with very good English skills. The tenses have been standardized, and places that are unclear to understand have been slightly changed.

We would like to thank the Reviewer very much for pointing out the weaknesses of our Review. We believe that after the corrections made, its value has increased and it is suitable for publication.

Best regards

Szymon Viscardi

Anna Duda-Madej

Reviewer 2 Report

Comments and Suggestions for Authors

In the manuscript “Novel Siderophore Cephalosporin and Combinations of Cephalosporins with β-Lactamase Inhibitors as an Advancement in Treatment of Ventilator Associated Pneumonia”, the study introduced the drugs for their clinical trials, in vitro studies, case studies, structures, and biological functions. The study provides interesting clinical and biological knowledge for the drugs with potential to overcome multidrug resistance. There are a few comments from this reviewer.

1.     Instead of stacking the clinical studies, in vitro studies, and case studies, I would suggest synthesizing information and summarize what we can learn from these studies, given that these studies were already listed in tables.

2.     In figure 5, instead of clarifying the components in the legend, I would suggest labelling them in the figure. This helps with visualization given there are many different components in the figure.

3.     Since beta-lactamase inhibitor is a focus of the paper, I would suggest discussing the molecular mechanisms earlier in the paper before going into each drug.

4.     In addition to listing the abbreviations, I would suggest including the full names and briefly discuss what they mean if necessary at the place they firstly appear in the paper.

Author Response

Dear Reviewer,

Thank you very much for studying the topics undertaken in our Article and for your detailed review. In correcting our Review, we followed all your recommendations, for which we are very thankful. Our corrections are presented below:

  1. Instead of stacking the clinical studies, in vitro studies, and case studies, I would suggest synthesizing information and summarize what we can learn from these studies, given that these studies were already listed in tables.

The studies have been listed and briefly summarized in tables, which makes it easier to compare the results with each other: the specific studies for a particular drug, and the combination of two different drugs described in the study (for example, to compare a particular aspect of their use). In our opinion, the detailed descriptions are necessary because they contain data relevant to a broader understanding of the topic that were difficult to include in a table. For example, in the sections on clinical trials, the tables included only the endpoints of the studies, rather than every parameter that is present in the description. Similarly, in the case of in vitro studies, the tables compare only MIC50 and MIC90 values, while the descriptions preceding these tables additionally include data on the percentage of activity of a given formulation against specific pathogens and their comparison with comparators. For this reason, the suggestion that all descriptive sections preceding the comparative tables in the in vitro studies and clinical trials sections should be removed seems unjustified. We have made every effort to make the descriptions so concise and integral to the corresponding tables that we do not see the point of creating a single summary of the data, which in our opinion, could be much less readable.

  1. In figure 5, instead of clarifying the components in the legend, I would suggest labelling them in the figure. This helps with visualization given there are many different components in the figure.

Full names have been applied to the Figure 5. We encourage the Reviewer to read the revised Figure. We hope the message is clearer now.

  1. Since beta-lactamase inhibitor is a focus of the paper, I would suggest discussing the molecular mechanisms earlier in the paper before going into each drug.

Thank you for this comment. The molecular mechanisms were transferred before the analysis of each drug.

  1. In addition to listing the abbreviations, I would suggest including the full names and briefly discuss what they mean if necessary at the place they firstly appear in the paper.

All abbreviations used have been checked to explain the abbreviations in the text. A list of abbreviations has been added additionally to maintain order and for a more complete understanding by the reader.

We would like to thank the Reviewer very much for pointing out the weaknesses of our Review. We believe that after the corrections made, its value has increased and it is suitable for publication.

Best regards

Szymon Viscardi

Anna Duda-Madej

Reviewer 3 Report

Comments and Suggestions for Authors

This paper offers an insightful review of innovative siderophore cephalosporins and their combinations with β-lactamase inhibitors for treating ventilator-associated pneumonia. The integration of in vitro and clinical study data lays a solid groundwork for appreciating the capabilities of these therapies.

Nevertheless, the paper could be enhanced by considering the following recommendations:

Incorporating in vivo studies could significantly bolster the research by delivering more substantial and relevant data regarding the efficacy and safety of the discussed antibiotics.

The order of sections—clinical studies, in vitro studies, and case studies—while workable, is not typically the most logical for structuring research content. A more conventional approach starts with in vitro studies, followed by in vivo studies, then clinical studies, and finally case studies.

Future Research Directions: Broadening the discussion to include future research avenues, particularly in the development of new drugs or alternative therapies, would be advantageous. This could cover topics such as ongoing clinical trials, emerging drug categories, or innovative therapeutic strategies.

Comments on the Quality of English Language

The quality of the English language used in the paper is generally good, ensuring that the content is clear and understandable. However, for optimal clarity and professionalism, a few areas could benefit from further refinement:

1.        Grammar and Syntax: Some sections might benefit from a review to correct minor grammatical errors and improve sentence structure. This could enhance the readability and flow of the paper.

2.        Precision and Conciseness: Aim to make language more precise and concise. Eliminating redundant phrases and refining verbose sentences can make the arguments more direct and impactful.

Fro example:

Original Sentence

"Due to the resistance developed by bacteria, it becomes hard to treat them."

Revised Sentence

"Due to the resistance that bacteria have developed, treating them has become challenging."

Example Changes

Clarity and Flow: Rearranging the sentence structure improves readability and maintains a formal tone.

Word Choice: Using "challenging" instead of "hard" fits better in a scientific context.

Author Response

Dear Reviewer,

Thank you very much for studying the topics undertaken in our Article and for your detailed review. In correcting our Review, we followed all your recommendations, for which we are very thankful. Our corrections are presented below:

Incorporating in vivo studies could significantly bolster the research by delivering more substantial and relevant data regarding the efficacy and safety of the discussed antibiotics.

Due to the fact that the study relates strictly to the issue of VABP treatment, we primarily included in the survey information on the treatment of this condition in patients (clinical trials and case studies) and the antimicrobial properties of specific formulations against in vitro pathogens. We feel that it is these studies that are most relevant to the potential audience of the review, including practicing physicians. There is a lack of in vivo studies addressing VABP as a disease entity. Given the complex pathomechanism of this disease, the risk factors in patients and the specifics of ICU hospitalization, we feel that an in vivo study cannot reflect the natural course of this condition. An in vivo study would not contribute anything beyond what is reported in clinical studies. For this reason, we have not included this type of study in the article.

The order of sections—clinical studies, in vitro studies, and case studies—while workable, is not typically the most logical for structuring research content. A more conventional approach starts with in vitro studies, followed by in vivo studies, then clinical studies, and finally case studies.

The order of the results presented has been changed in accordance with the Reviewer's comments. We invite the Reviewer to review the new arrangement of our Manuscript.

Future Research Directions: Broadening the discussion to include future research avenues, particularly in the development of new drugs or alternative therapies, would be advantageous. This could cover topics such as ongoing clinical trials, emerging drug categories, or innovative therapeutic strategies.

The conclusion has been expanded to include information on future research directions. This section now reads as follows:

„Limited treatment options for patients developing VABP should also encourage the search for innovative treatments for this life-threatening infection. Due to the increase in antibiotic resistance, it is necessary to develop products that can combine the properties of classic drugs (e.g. inhibition of PBP protein) with a novel mechanism to break down antibiotic resistance (e.g. siderophore mechanism). Future directions of research should oscillate around the search for new ways of influencing bacterial cells. High hopes in this field may be raised by substances of natural origin (e.g. plant extracts and alkaloids), which due to their cytostatic properties have already been used in, among others, the therapy of oncological diseases. It also seems particularly important to search for new combinations of classic antibiotics with antibacterial substances of natural origin. Detection of additive or synergistic interactions between these compounds could reduce the use of last resort drugs (colistin) in favour of preparations with, among others, lower toxicity. Although more research is needed, the combination of innovative therapeutic strategies and rapid molecular diagnostics systems (MALDI-TOF MS) is a worthwhile option in the face of the growing global problem of antibiotic resistance”.

In addition, the Manuscript was reviewed by a person with very good English skills. The tenses have been standardized, and places that are unclear to understand have been slightly changed.

We would like to thank the Reviewer very much for pointing out the weaknesses of our Review. We believe that after the corrections made, its value has increased and it is suitable for publication.

Best regards

Szymon Viscardi

Anna Duda-Madej

Round 2

Reviewer 2 Report

Comments and Suggestions for Authors

All the comments were addressed. 

Reviewer 3 Report

Comments and Suggestions for Authors

the author has addressed all my concerns.